# Concept-Based Unsupervised Domain Adaptation

Xinyue Xu [* 1]   Yueying Hu [* 1]   Hui Tang [1]   Yi Qin [1]   Lu Mi [2]   Hao Wang [3]   Xiaomeng Li [1]

## Abstract

Concept Bottleneck Models (CBMs) enhance interpretability by explaining predictions through human-understandable concepts but typically assume that training and test data share the same distribution. This assumption often fails under domain shifts, leading to degraded performance and poor generalization. To address these limitations and improve the robustness of CBMs, we propose the **Concept-based Unsupervised Domain Adaptation (CUDA)** framework. CUDA is designed to: (1) align concept representations across domains using adversarial training, (2) introduce a relaxation threshold to allow minor domain-specific differences in concept distributions, thereby preventing performance drop due to over-constraints of these distributions, (3) infer concepts directly in the target domain without requiring labeled concept data, enabling CBMs to adapt to diverse domains, and (4) integrate concept learning into conventional domain adaptation (DA) with theoretical guarantees, improving interpretability and establishing new benchmarks for DA. Experiments demonstrate that our approach significantly outperforms the state-of-the-art CBM and DA methods on real-world datasets.

## 1. Introduction

Black-box models often lack interpretability, making them difficult to trust in high-stakes scenarios. Concept Bottleneck Models (CBMs) (Koh et al., 2020; Ghorbani et al., 2019) tackle this interpretability issue by using human-understandable concepts. These models first predict concepts from the input data and then use concepts to predict the final label, thereby improving their interpretability, e.g., predicting concepts "black eyes" and "solid belly" to classify and interpret the bird species "Sooty Albatross". This also

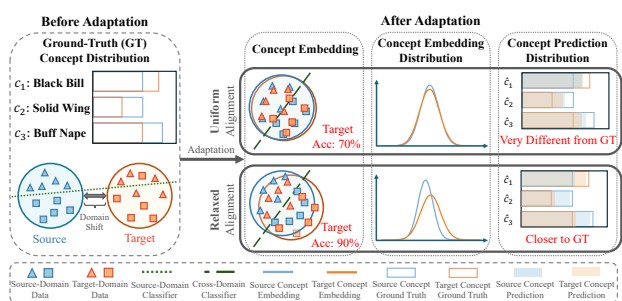

*Figure 1.* Illustration of our key idea. **Left:** Ground-truth (GT) concept distributions (for each concept) (**top**) and data distributions (**bottom**). **Right:** Uniform alignment (**top**) and relaxed alignment (**bottom**) after adaptation. Our relaxed alignment allows for greater differences between source and target concept distributions; such flexibility leads to predicted concept distributions closer to the ground truth and therefore higher final classification accuracy.

allows experts to understand misclassifications and make necessary interventions when needed (Abid et al., 2022). However, existing CBMs typically assume that the training and test data share the same distribution, which limits their effectiveness in real-world applications where domain shifts between training and test sets are common. For example, methods such as CBMs (Koh et al., 2020) and Concept Embedding Models (Zarlenga et al., 2022) demonstrate a significant drop in performance when tested under domain shift conditions. These models achieve only around 66% accuracy under background shifts, a notable drop compared to their 80% accuracy on test sets that align with the training distribution, as observed on the CUB dataset (Wah et al., 2011) (Sec. 5). Despite these findings, the challenge of designing interpretable models capable of handling real-world domain shifts remains largely underexplored.

A straightforward approach is to combine CBMs with Domain Adaptation (DA) (Ben-David et al., 2010; Ganin et al., 2016), which tackles domain shifts by utilizing labeled data from source domains alongside unlabeled (or sparsely labeled) data from target domains. Specifically, a naive combination of CBMs and DA would simply add concept learning into DA models. Unfortunately, this method performs poorly (more results in Appendix C.2) for two reasons. First, it enforces separate class-wise and concept-wise alignment, failing to unify them into a single feature space, limiting both interpretability and generalization. Second, existing DA methods assume uniform (perfect) alignment between

---

[*]Equal contribution [1]The Hong Kong University of Science and Technology [2]Georgia Institute of Technology [3]Rutgers University. Correspondence to: Xiaomeng Li <eexmli@ust.hk>.

*Proceedings of the 42$^{st}$ International Conference on Machine Learning*, Vancouver, Canada. PMLR 267, 2025. Copyright 2025 by the author(s).

source and target concepts, overlooking domain-specific variations that are essential for CBMs to capture meaningful and interpretable concepts. As shown in Fig. 1 (upper-right), while uniform (perfect) alignment can strictly align source and target concepts, it overlooks the inherent differences between concepts across domains. Such over-constraints lead to significant performance drops.

One of our key ideas is therefore to introduce a degree of relaxation. As shown in Fig. 1 (bottom-right), our relaxed alignment allows for greater differences between source and target concept distributions, e.g., allowing the proportion of the concept "Primary Color: Brown" to be 19% in the source domain and 17% in the targe domain for bird classification; such flexibility leads to predicted concept distributions closer to the ground truth and therefore higher final classification accuracy. Specifically, we propose a novel **Concept-based Unsupervised Domain Adaptation (CUDA)** framework, a simple yet effective approach with strong generalization capabilities. To achieve this, we introduce a novel relaxed uniform alignment loss that adapts more flexibly across domains. This approach enables the learning of domain-invariant concept embeddings while effectively preserving domain-specific variations. We summarize our contributions as follows:

- We provide the first generalization error bound for CBMs, with theoretical analysis on how concept embeddings can be utilized to align source and target distributions in DA.
- Inspired by the theoretical analysis, we propose the first general framework for concept-based DA, providing both cross-domain generalization and concept-based interpretability.
- We improve generalization of CBMs and eliminate the need for labeled concept data and retraining on the target domain, enabling adaptation to diverse domains.
- Experiments on real-world datasets show that our method significantly outperforms state-of-the-art CBM and DA models, establishing new benchmarks for concept-based domain adaptation.

## 2. Related Work

**Concept Bottleneck Models (CBMs)** (Koh et al., 2020) use bottleneck models to map inputs into the concept space and make predictions based on the extracted concepts. Concept Embedding Models (CEMs) (Zarlenga et al., 2022) improve performance by using a weighted mixture of positive and negative embeddings for each concept. Energy-based Concept Bottleneck Models (ECBMs) (Xu et al., 2024) unify prediction, concept correction, and interpretation as conditional probabilities under a joint energy formulation. Post-hoc Concept Bottleneck Models (PCBMs) (Yuksekgonul et al., 2022) employ a post-hoc explanation model with resid-

ual fitting, storing Concept Activation Vectors (CAVs) (Kim et al., 2018) in a concept bank, which eliminates the need for retraining on target domains. DISC (Wu et al., 2023) complements this by building a comprehensive concept bank that covers potential spurious concept candidates. CONDA (Choi et al., 2024) further extends PCBMs by performing test-time adaptation using pseudo-labels generated by foundation models. Our approach combines the advantages of these methods: it requires neither retraining nor concept labels in the target domain, while retaining the complete interpretability of the original concepts. Unlike PCBMs and CONDA, our method supports direct evaluation of concept learning performance, ensuring both interpretability and strong performance in the target domain. Note that our work is orthogonal to unsupervised concept interpretation of foundation models (Wang et al., 2024a;b; Wang & Yeung, 2016; 2020).

**Domain Adaptation.** In domain adaptation, the task remains the same across source and target domains, while the data distributions differ across domains (Pan & Yang, 2009). Our work assumes unlabeled data in the target domain, falling under the category of unsupervised domain adaptation (UDA) (Beijbom, 2012). Existing UDA methods primarily focus on learning domain-invariant features, enabling a classifier trained on source to be applied to target data. These methods can be broadly categorized into three adaptation paradigms: input-level (Sankaranarayanan et al., 2018; Hoffman et al., 2018), feature-level (Ganin et al., 2016; Saito et al., 2018; Xu et al., 2022; Liu et al., 2023; Xu et al., 2023; Huang et al., 2024), and output-level (Zhang et al., 2019b; Tang et al., 2020; Hu et al., 2022). Input-level adaptation stylizes data (e.g., images) from one domain to match the style of another. This involves generating source-like target data as regularization (Sankaranarayanan et al., 2018) or target-like source data as training data (Hoffman et al., 2018), often using GANs (Goodfellow et al., 2014). Feature-level adaptation minimizes feature distribution discrepancies between domains (Long et al., 2015) or employs adversarial training at the domain (Ganin et al., 2016; Xu et al., 2023) or class levels (Saito et al., 2018; Huang et al., 2024). Output-level adaptation focuses on learning target-discriminative features through self-training with pseudo-labels (Zhang et al., 2019b; Tang et al., 2020; Hu et al., 2022). None of the methods above provide concept-level interpretability. In contrast, our approach, for the first time, introduces the concept-level perspective for adaptation. By leveraging concept learning, we bridge domain discrepancies while achieving concept-based interpretable UDA.

## 3. Methodology

In this section, we begin by analyzing the generalization error bound for CBMs and then discuss our proposed method

inspired by the analysis. A detailed theoretical analysis is provided in Sec. 4.

**Problem Setting and Notations.** We consider the concept-based UDA setting with $Q$ classes and $K$ concepts. The input, label, and concepts are denoted as $\boldsymbol{x} \in \mathcal{X}$, $\boldsymbol{y} \in \mathcal{Y} \subset \{0,1\}^Q$, and $\boldsymbol{c} \in \mathcal{C} = \{0,1\}^K$, respectively; note that $\mathcal{Y}$ represents the space of $Q$-dimensional one-hot vectors while $\mathcal{C}$ does not. We use discrete domain indices (Wang et al., 2020) $u = 0$ and $u = 1$ to denote source and target domains, respectively. Given the labeled data $\{(\boldsymbol{x}_i^s, \boldsymbol{y}_i^s, \boldsymbol{c}_i^s)\}_{i=1}^n$ from source domain ($u = 0$), and unlabeled data $\{\boldsymbol{x}_i^t\}_{i=1}^m$ from target domain ($u = 1$), the goal is to accurately predict both the classification labels $\{\boldsymbol{y}_i^t\}_{i=1}^m$ and the unlabeled concepts $\{\boldsymbol{c}_i^t\}_{i=1}^m$ in the target domain.

### 3.1. Generalization Error Bound for CBMs

Previous works on CBMs have primarily been evaluated on background shift tasks (Koh et al., 2020), but they lack theoretical analysis of the generalization error bound. To address this limitation and provide deeper insights into our proposed method, we begin by analyzing the generalization error bound for CBMs. Although our primary focus is on binary classification, our framework can extend to multi-class classification following Zhang et al. (2019a; 2020), which we leave for future work.

**Generalization Bound without Concept Terms.** Building on the framework established in Ben-David et al. (2006; 2010), we formalize the data generation process for both source domain and target domain using marginal (data) distribution and underlying labeling function pairs, denoted as $\langle \mathcal{D}_S, f_S \rangle$ for the source domain and $\langle \mathcal{D}_T, f_T \rangle$ for the target domain. Here, $\mathcal{D}_S$ and $\mathcal{D}_T$ denote the marginal distributions over the input space $\mathcal{X}$, while $f_S : \mathcal{X} \to [0,1]$ and $f_T : \mathcal{X} \to [0,1]$ represent the labeling functions that assign the probability of an instance being classified as label 1 in the source and target domains, respectively. We adopt a concept embedding encoder $E : \mathcal{X} \to \mathcal{V} \subset \mathbb{R}^J$, a function which maps inputs to concept embeddings. This induces distributions $\widetilde{D}_S$ and $\widetilde{D}_T$ over the concept embedding space $\mathcal{V}$, as well as corresponding labeling functions:

$$\widetilde{f}_S(\boldsymbol{v}) \triangleq \mathbb{E}_{\boldsymbol{x} \sim \mathcal{D}_S}[f_S(\boldsymbol{x}) \mid E(\boldsymbol{x}) = \boldsymbol{v}],$$
$$\widetilde{f}_T(\boldsymbol{v}) \triangleq \mathbb{E}_{\boldsymbol{x} \sim \mathcal{D}_T}[f_T(\boldsymbol{x}) \mid E(\boldsymbol{x}) = \boldsymbol{v}].$$

We define a hypothesis $h : \mathcal{V} \to [0,1]$ as a predictor operating over the concept embedding space $\mathcal{V}$. For any embedding $\boldsymbol{v} \in \mathcal{V}$, $h(\boldsymbol{v})$ outputs the predicted probability that the classification label is 1. The error of $h$ on the source and target domains is then defined as:

$$\epsilon_S(h) \triangleq \epsilon_S(h, \widetilde{f}_S) = \mathbb{E}_{\boldsymbol{v} \sim \widetilde{\mathcal{D}}_S}\left[\left|\widetilde{f}_S(\boldsymbol{v}) - h(\boldsymbol{v})\right|\right],$$
$$\epsilon_T(h) \triangleq \epsilon_T(h, \widetilde{f}_T) = \mathbb{E}_{\boldsymbol{v} \sim \widetilde{\mathcal{D}}_T}\left[\left|\widetilde{f}_T(\boldsymbol{v}) - h(\boldsymbol{v})\right|\right].$$

For any $h \in \mathcal{H}$ with $\mathcal{H}$ as the hypothesis space, Ben-David et al. (2006; 2010) present a theoretical upper bound on the target error $\epsilon_T(h)$:

$$\epsilon_T(h) \leq \epsilon_S(h) + \tfrac{1}{2}d_{\mathcal{H}\Delta\mathcal{H}}(\widetilde{D}_S, \widetilde{D}_T) + \eta, \quad (1)$$

where $\eta = \min_{h \in \mathcal{H}} (\epsilon_S(h) + \epsilon_T(h))$ denotes the error of a joint ideal hypothesis on both source and target domains, and the $\mathcal{H}\Delta\mathcal{H}$ divergence $d_{\mathcal{H}\Delta\mathcal{H}}(\widetilde{D}_S, \widetilde{D}_T)$ represents the worst-case source-target domain discrepancy over *concept embedding space* (different from Ben-David et al. (2010), which is in the input space).

**Concept Embeddings $\boldsymbol{v}_i$.** Given that using scalar representations for concepts can significantly degrade predictive performance in realistic settings (Mahinpei et al., 2021; Dominici et al., 2024), we choose to use a more robust approach that constructs positive and negative semantic embeddings for each concept (Zarlenga et al., 2022; Xu et al., 2024). Specifically, the concept embedding $\boldsymbol{v}$ is represented as a concatenation of sub-embeddings for $K$ concepts, i.e. $\boldsymbol{v} = [\boldsymbol{v}_i]_{i=1}^K \in \mathbb{R}^J$, where each sub-embedding $\boldsymbol{v}_i$ is a combination of its positive and negative embeddings weighted by the predicted concept probability $\widehat{c}_i$

$$\boldsymbol{v}_i \triangleq \widehat{c}_i \cdot \boldsymbol{v}_i^{(+)} + (1 - \widehat{c}_i) \cdot \boldsymbol{v}_i^{(-)}, \quad (2)$$

where $\widehat{\boldsymbol{c}} = [\widehat{c}_i]_{i=1}^K \in \mathbb{R}^K$.

**Ideal Concept Embeddings $\boldsymbol{v}_i^c$.** Note that ground-truth concepts $\boldsymbol{c} = [c_i]_{i=1}^K \in \{0,1\}^K$ are only accessible in the source domain, which allows us to define an idealized scenario for analyzing the source error. In this scenario, we replace the predicted concept probabilities $\widehat{\boldsymbol{c}}$ with the ground-truth concepts $\boldsymbol{c}$ to construct the ideal concept embeddings $\boldsymbol{v}^c = [\boldsymbol{v}_i^c]_{i=1}^K \in \mathbb{R}^J$, with each $\boldsymbol{v}_i^c$ defined as:

$$\boldsymbol{v}_i^c \triangleq c_i \cdot \boldsymbol{v}_i^{(+)} + (1 - c_i) \cdot \boldsymbol{v}_i^{(-)},$$

where $c_i$ denotes the ground truth of the $i$-th concept. This eliminates the noise introduced by the prediction, providing a minimal-error baseline that isolates the inherent limitations of the model itself.

**Source Error with Ideal Concept Embeddings.** To quantify performance under this noise-free baseline, we define the source error for $\boldsymbol{v}^c$:

$$\epsilon_S^c(h) \triangleq \epsilon_S^c(h, \widetilde{f}_S^c) = \mathbb{E}_{\boldsymbol{v}^c \sim \widetilde{\mathcal{D}}_S^c}\left[\left|\widetilde{f}_S^c(\boldsymbol{v}^c) - h(\boldsymbol{v}^c)\right|\right],$$

where $\widetilde{\mathcal{D}}_S^c$ denotes the marginal distribution over $\boldsymbol{v}^c$, and $\widetilde{f}_S^c$ is the corresponding induced labeling function, defined as:

$$\widetilde{f}_S^c(\boldsymbol{v}^c) \triangleq \mathbb{E}_{\boldsymbol{x} \sim \mathcal{D}_S}[f_S(\boldsymbol{x}) \mid E(\boldsymbol{x}) = \boldsymbol{v}^c].$$

**Generalization Bound with Concept Terms.** With this setup, we are ready to perform a generalization error analysis of concept-based models for the binary classification task. A complete proof can be found in Appendix B.1.

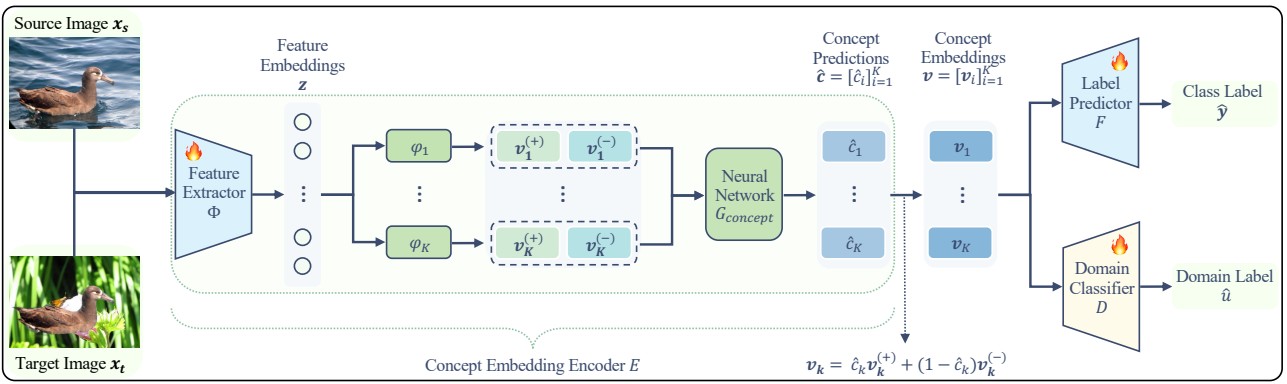

*Figure 2.* Overview of our CUDA framework. The framework takes source and target domain images as inputs to first learn feature embeddings. Positive embeddings $\boldsymbol{v}_i^{(+)}$ and negative embeddings $\boldsymbol{v}_i^{(-)}$ are then derived from these feature embeddings. These are passed through the neural network $G_{concept}$ to obtain concept predictions $\widehat{\boldsymbol{c}}$, which are subsequently combined to construct the final concept embeddings $\boldsymbol{v}$. During training, adversarial training is employed: the domain classifier (discriminator) is trained first, followed by the concept embedding encoder and label predictor. These two steps are alternated throughout the training process.

By comparing the noise-free source error $\epsilon_S^{\mathbf{c}}$ (which serves as the theoretical baseline for evaluating performance under ideal conditions) with the actual source error $\epsilon_S$ that incorporates noisy predicted probabilities, we can directly quantify the additional error introduced by prediction noise. This relationship is formalized in the following lemma.

**Lemma 3.1 (Source Error with Predicted Concept Embeddings).** *Let $\mathcal{H}$ be a hypothesis space where all hypotheses $h \in \mathcal{H}$ are $L$-Lipschitz continuous under the Euclidean norm $\|\cdot\|_2$ for some constant $L > 0$. Assume that for all $\boldsymbol{v} \in \mathcal{V}$, $\|\boldsymbol{v}\|_2$ is bounded. Then, for any $h_1, h_2 \in \mathcal{H}$, there exists a finite constant $r > 0$ such that*

$$\epsilon_S(h_1, h_2) \leq \epsilon_S^{\mathbf{c}}(h_1, h_2) + r \cdot \mathbb{E}_S\left[\|\widehat{\boldsymbol{c}} - \boldsymbol{c}\|_2\right],$$

*where $\epsilon_S(h_1, h_2) = \mathbb{E}_{\boldsymbol{v} \sim \widetilde{\mathcal{D}}_S}\left[|h_1(\boldsymbol{v}) - h_2(\boldsymbol{v})|\right]$ and $\epsilon_S^{\mathbf{c}}(h_1, h_2) = \mathbb{E}_{\boldsymbol{v}^{\mathbf{c}} \sim \widetilde{\mathcal{D}}_S^{\mathbf{c}}}\left[|h_1(\boldsymbol{v}^{\mathbf{c}}) - h_2(\boldsymbol{v}^{\mathbf{c}})|\right]$ are the disagreement between hypotheses $h_1$ and $h_2$ w.r.t. distributions $\widetilde{\mathcal{D}}_S$ and $\widetilde{\mathcal{D}}_S^{\mathbf{c}}$, respectively, and $\mathbb{E}_S$ denotes the expectation taken over the source distribution.*

Lemma 3.1 quantitatively connects the concept prediction performance to the source error. Specifically, $\mathbb{E}_S\left[\|\widehat{\boldsymbol{c}} - \boldsymbol{c}\|_2\right]$ quantifies the discrepancy between the predicted concepts $\widehat{\boldsymbol{c}}$ and ground-truth concepts $\boldsymbol{c}$, serving as a measure of the accuracy of concept prediction. We defer the discussion of the validity of the $L$-Lipschitz continuity assumption to Appendix C.2. With this foundation, we are now ready to derive a bound on the target error for concept-based models.

**Theorem 3.1 (Target-Domain Error Bound for Concept-Based Models).** *Under the assumption of Lemma 3.1, for any $h \in \mathcal{H}$, we have:*

$$\epsilon_T(h) \leq \epsilon_S^{\mathbf{c}}(h) + \tfrac{1}{2}d_{\mathcal{H}\Delta\mathcal{H}}\left(\widetilde{\mathcal{D}}_S^{\mathbf{c}}, \widetilde{\mathcal{D}}_T\right) + \eta^{\mathbf{c}} \quad (3)$$
$$+ R \cdot \mathbb{E}_S\left[\|\widehat{\boldsymbol{c}} - \boldsymbol{c}\|_2\right],$$

*where $R > 0$ is a finite constant, $\eta^{\mathbf{c}} = \min\limits_{h \in \mathcal{H}} \epsilon_S^{\mathbf{c}}(h) + \epsilon_T(h)$, and $d_{\mathcal{H}\triangle\mathcal{H}}(\widetilde{\mathcal{D}}_S^{\mathbf{c}}, \widetilde{\mathcal{D}}_T)$ denotes the $\mathcal{H}\Delta\mathcal{H}$ divergence between distribution $\widetilde{\mathcal{D}}_S^{\mathbf{c}}$ and distribution $\widetilde{\mathcal{D}}_T$.*

Theorem 3.1 implies that the target error $\epsilon_T$ can be minimized by reducing the source error with ground-truth concepts $\epsilon_S^{\mathbf{c}}$, the $\mathcal{H}\Delta\mathcal{H}$ divergence $d_{\mathcal{H}\triangle\mathcal{H}}(\widetilde{\mathcal{D}}_S^{\mathbf{c}}, \widetilde{\mathcal{D}}_T)$, and the discrepancy $\mathbb{E}_S\left[\|\widehat{\boldsymbol{c}} - \boldsymbol{c}\|_2\right]$ simultaneously, thereby achieving high classification accuracy on the target domain.

### 3.2. Concept-Based Unsupervised Domain Adaptation

Inspired by Theorem 3.1, we propose a game-theoretic framework, dubbed **C**oncept-based **U**nsupervised **D**omain **A**daptation (CUDA). Fig. 2 provides an overview of CUDA, which involves four players:

- a **concept embedding encoder** $E$ which generates the concept embedding $\boldsymbol{v} = E(\boldsymbol{x})$ given the input $\boldsymbol{x}$,
- a **concept probability encoder** $E_{prob}$ which predicts concepts $\widehat{\boldsymbol{c}} = E_{prob}(\boldsymbol{x})$ (though $E_{prob}$ is part of $E$, we treat them separately for analysis purposes),
- a **discriminator** $D$ which identifies the domain $\widehat{u}$ using the concept embedding $\boldsymbol{v}$, i.e. $\widehat{u} = D(\boldsymbol{v})$, and
- a **predictor** $F$ which predicts the classification label $\widehat{\boldsymbol{y}}$ based on the concept embedding $\widehat{\boldsymbol{y}} = F(\boldsymbol{v})$.

**The Need for Relaxed Alignment.** Before introducing the game, note that the adversarial interaction between $E$ and $D$ forces $E$ to strip all domain-specific information from the concept embedding $\boldsymbol{v}$ at the optimal point, making $\boldsymbol{v}$ effectively domain-invariant. Intuitively, since the concept probability $\widehat{\boldsymbol{c}}$ is part of $\boldsymbol{v}$, $\widehat{\boldsymbol{c}}$ should also become domain-invariant, achieving perfect (uniform) alignment across domains. However, the concepts in the source and target domains are often inconsistent due to differences in data distributions in practice (Xu et al., 2022; Liu et al., 2023);

such discrepancies make the uniform alignment overly restrictive, as it may impose unnecessary constraints on $\widehat{c}$, therefore harming performance in the target domain. To address this gap, we draw inspiration from (Xu et al., 2022; Liu et al., 2023) and propose a relaxed alignment mechanism on $\boldsymbol{v}$, which naturally translates to tolerating smaller discrepancies in $\widehat{c}$ between the source and target domains.

**Overall Objective Function.** Formally, CUDA solves the following optimization problem:

$$\min_{D} \mathcal{L}_d(E, D), \qquad (4)$$

$$\min_{E, E_{prob}, F} \mathcal{L}_p(E, F) + \lambda_c \mathcal{L}_c(E_{prob}) - \lambda_d \widetilde{\mathcal{L}}_d(E, D), \quad (5)$$

where $\mathcal{L}_p$ is the prediction loss, $\widetilde{\mathcal{L}}_d$ and $\mathcal{L}_d$ are the discriminator loss *with* and *without relaxation*, respectively (more details below), and $\mathcal{L}_c$ is the concept loss. The hyperparameters $\lambda_d$ and $\lambda_c$ balance $\mathcal{L}_p(E, F)$, $\mathcal{L}_c(E_{prob})$ and $\widetilde{\mathcal{L}}_d(E, D)$. Below, we discuss each term in detail.

**Prediction Loss $\mathcal{L}_p$ and Predictor $F$.** The prediction loss $\mathcal{L}_p(E, F)$ in Eqn. 5 is defined as:

$$\mathcal{L}_p(E, F) \triangleq \mathbb{E}_S \left[ L_p(F(E(\boldsymbol{x})), \boldsymbol{y}) \right], \qquad (6)$$

where $L_p$ is the cross-entropy loss, $F(E(\boldsymbol{x})) \in \mathbb{R}^Q$ and each element $F(E(\boldsymbol{x}))_i$ is the predicted probability for class $i$, and $\mathbb{E}_S$ denotes the expectation taken over the source data distribution $p_S(\boldsymbol{x}, \boldsymbol{y}, \boldsymbol{c})$; note that the label $\boldsymbol{y}$ and ground-truth concepts $\boldsymbol{c}$ are only accessible in the source domain.

**Concept Embedding Encoder $E$.** The concept embedding encoder $E$ generates both concept predictions $\widehat{c}$ and concept embeddings $\boldsymbol{v}$. As presented in Fig. 2, positive and negative embeddings for the $i$-th concept are firstly constructed as: $[\boldsymbol{v}_i^{(+)}, \boldsymbol{v}_i^{(-)}] = \varphi_i(\Phi(\boldsymbol{x}))$, where $\Phi(\cdot)$ is a pretrained backbone and $\varphi_i(\cdot)$ is the linear layer. Then the concatenated embeddings $[\boldsymbol{v}_i^{(+)}, \boldsymbol{v}_i^{(-)}]$ are passed through $G_{concept}$ to predict the concept probability: $\widehat{c}_i = G_{concept}([\boldsymbol{v}_i^{(+)}, \boldsymbol{v}_i^{(-)}])$. Thus, we have:

$$\widehat{\boldsymbol{c}} = E_{prob}(\boldsymbol{x}) = [G_{concept}([\boldsymbol{v}_i^{(+)}, \boldsymbol{v}_i^{(-)}])]_{i=1}^K,$$

where $E_{prob}(\cdot)$ is the concept probability encoder composing $\Phi(\cdot)$, $\varphi(\cdot)$ and $G_{concept}(\cdot)$.

As mentioned in Eqn. 2, we then use the full concept embedding encoder $E$ to compute the concept embedding $\boldsymbol{v}$:

$$\boldsymbol{v} = E(\boldsymbol{x}) = [\boldsymbol{v}_i]_{i=1}^K = [\widehat{c}_i \cdot \boldsymbol{v}_i^{(+)} + (1 - \widehat{c}_i) \cdot \boldsymbol{v}_i^{(-)}]_{i=1}^K$$
$$= [(E_{prob}(\boldsymbol{x}))_i \cdot \boldsymbol{v}_i^{(+)} + (1 - (E_{prob}(\boldsymbol{x}))_i) \cdot \boldsymbol{v}_i^{(-)}]_{i=1}^K.$$

Note that the concept probability encoder $E_{prob}$ is part of the full concept embedding encoder $E$. We separate concept probability encoder $E_{prob}$ out to facilitate theoretical analysis. Specifically, $E_{prob}$ is optimized to minimize

$\mathbb{E}_S [\|\widehat{c} - c\|_2]$, ensuring accurate concept probability estimation. Meanwhile, $E$ collaborates with the predictor $F$ to reduce the source error $\epsilon_S^c$, and "fools" the discriminator $D$ to minimize the $\mathcal{H}\Delta\mathcal{H}$ divergence $d_{\mathcal{H}\Delta\mathcal{H}}(\widetilde{D}_S^c, \widetilde{D}_T)$. Together, they jointly optimize the upper bound of the target-domain error, i.e., Eqn. 3 of Theorem 3.1.

**Concept Loss $\mathcal{L}_c$.** In Eqn. 5, the concept loss is defined as:

$$\mathcal{L}_c(E_{prob}) \triangleq \mathbb{E}_S \left[ L_c(E_{prob}(\boldsymbol{x}), \boldsymbol{c}) \right], \qquad (7)$$

where $L_c$ is the binary cross-entropy loss, $E_{prob}(\boldsymbol{x}) \in \mathbb{R}^K$, where each dimension $(E_{prob}(\boldsymbol{x}))_i$ is the predicted concept probability for concept $i$; the corresponding ground-truth concept is $c_i$ (note that $\boldsymbol{c} = [c_i]_{i=1}^K \in \mathbb{R}^K$).

**Discriminator Loss without Relaxation $\mathcal{L}_d$ and Discriminator $D$.** The discriminator $D$ identifies the domain $u$ from the concept embedding $\boldsymbol{v}$. Given $E$, the discriminator loss

$$\mathcal{L}_d(E, D) \triangleq \mathbb{E} \left[ L_d(D(E(\boldsymbol{x})), u) \right], \qquad (8)$$

where $L_d$ is the binary cross-entropy loss, $u$ is the domain label which indicates whether $\boldsymbol{x}$ comes from the source ($u = 0$) or target ($u = 1$) domain, $\mathbb{E}$ denotes the expectation taken over the entire data distribution $p(\boldsymbol{x}, u)$, and $D(E(\boldsymbol{x}))$ denotes the probability of $\boldsymbol{x}$ belonging to the target domain.

**Relaxed Discriminator Loss $\widetilde{\mathcal{L}}_d$.** $\mathcal{L}_d$ is only used to learn the discriminator $D$ (Eqn. 4). To learn the encoder $E$ in Eqn. 5, we introduce a relaxed discriminator loss:

$$\widetilde{\mathcal{L}}_d(E, D) \triangleq \min \{\mathcal{L}_d(E, D), \tau\}, \qquad (9)$$

where $0 < \tau \leq \max \mathcal{L}_d(E, D)$ is a relaxation threshold, effectively controlling the tolerance for domain discrepancies in the concept embedding $\boldsymbol{v}$.

**Relaxed Discriminator Loss for Relaxed Alignment.** By capping the domain classification loss at $\tau$, this relaxation intentionally sacrifices a small amount of domain alignment, corresponding to the second term $\frac{1}{2} d_{\mathcal{H}\Delta\mathcal{H}}(\widetilde{D}_S^c, \widetilde{D}_T)$ of Eqn. 3, to reduce the concept prediction error in the fourth term $\mathbb{E}_S [\|\widehat{c} - c\|_2]$ of Eqn. 3. This trade-off enables a more flexible optimization of the concept embedding encoder $E$, balancing domain alignment and concept prediction accuracy. Besides, it allows the encoder $E$ to retain domain-specific information stemming from intrinsic differences between source and target concepts, crucial for downstream tasks (see Sec. 4 for a comprehensive analysis). We summarize CUDA's training procedure in Algorithm 1 of Appendix C.3. Essentially, it alternates between Eqn. 4 and 5 with adversarial training using Eqn. 6~9.

## 4. Theoretical Analysis for CUDA

In this section, we provide the theoretical guarantees for CUDA. All proofs are provided in Appendix B.2.

**Simplified Game.** We start by analyzing a simplified game which does not involve the concept probability encoder $E_{prob}$ and the predictor $F$. Specifically, we focus on

$$\min_D \mathcal{L}_d(E, D), \tag{10}$$

$$\max_E \widetilde{\mathcal{L}}_d(E, D) \triangleq \min\{\mathcal{L}_d(E, D), \tau\}, \tag{11}$$

where the discriminator loss without relaxation $\mathcal{L}_d$ is defined in Eqn. 8, and $0 < \tau \leq \max \mathcal{L}_d(E, D)$ is a relaxation threshold that quantifies the allowed deviation from uniform alignment of $\boldsymbol{v}$. Solving this game ensures that $D$ learns to distinguish domain representations, while $E$ can "fool" the discriminator with the relaxation threshold $\tau$, thereby flexibly aligning concept embeddings across domains.

Lemma 4.1 below analyzes the optimal discriminator $D$ in Eqn. 10 with the concept embedding encoder $E$ fixed.

**Lemma 4.1** (**Optimal Discriminator**). *For $E$ fixed, the optimal discriminator $D$ is*

$$D_E^*(\boldsymbol{v}) = \frac{p_T^{\boldsymbol{v}}(\boldsymbol{v})}{p_S^{\boldsymbol{v}}(\boldsymbol{v}) + p_T^{\boldsymbol{v}}(\boldsymbol{v})},$$

*where $p_S^{\boldsymbol{v}}(\boldsymbol{v})$ and $p_T^{\boldsymbol{v}}(\boldsymbol{v})$ are the probability density function of $\boldsymbol{v}$ in source and target domains, respectively.*

**Analyzing the Relaxed Discriminator Loss.** Given the optimal discriminator $D_E^*$ in Lemma 4.1, we define the relaxed discriminator objective in Eqn. 11 as:

$$\begin{aligned}\widetilde{C}_d(E) &\triangleq \widetilde{\mathcal{L}}_d(E, D_E^*) \\ &= \min\{\mathcal{L}_d(E, D_E^*), \tau\} = \min\{C_d(E), \tau\},\end{aligned} \tag{12}$$

where $C_d(E) \triangleq \mathcal{L}_d(E, D_E^*)$. Theorem 4.1 below shows that the global optimum of the game in Eqn. 10~11 corresponds to *relaxed alignment* of concept embeddings $\boldsymbol{v}$ and concept predictions $\widehat{\boldsymbol{c}}$ between source and target domains.

**Theorem 4.1** (**Relaxed Alignment**). *If the discriminator $D$ have enough capacity to be trained to reach optimum, the relaxed optimization objective $\widetilde{C}_d(E)$ defined in Eqn. 12 achieves its global maximum if and only if the concept embedding encoder satisfies the following conditions:*

$$\text{JSD}(p_S^{\boldsymbol{v}}(\boldsymbol{v}) \| p_T^{\boldsymbol{v}}(\boldsymbol{v})) = \log 2 - \tau, \tag{13}$$

$$\text{JSD}(p_S^{\widehat{\boldsymbol{c}}}(\widehat{\boldsymbol{c}}) \| p_T^{\widehat{\boldsymbol{c}}}(\widehat{\boldsymbol{c}})) = \log 2 - \tau - I(\boldsymbol{v}, u | \widehat{\boldsymbol{c}}), \tag{14}$$

*where $I(\cdot, \cdot | \cdot)$ is the conditional mutual information, $p_S^{\widehat{\boldsymbol{c}}}(\widehat{\boldsymbol{c}})$ and $p_T^{\widehat{\boldsymbol{c}}}(\widehat{\boldsymbol{c}})$ are the probability density function of $\widehat{\boldsymbol{c}}$ in source and target domains, respectively.*

Theorem 4.1 links the relaxation threshold $\tau$ in CUDA to the alignment of concept embedding $\boldsymbol{v}$'s distributions and concept prediction $\widehat{\boldsymbol{c}}$'s distributions across domains:

- When $\tau \in (0, \log 2)$, CUDA achieves **relaxed alignment**, and the degree of relaxation for $\widehat{\boldsymbol{c}}$ is guaranteed to be no greater than that of $\boldsymbol{v}$.

- When $\tau = \log 2$, CUDA achieves **uniform alignment**, which is defined in Definition 4.1 below.

**Definition 4.1** (**Uniform Alignment**). A concept-based DA model achieves uniform alignment if its encoder satisfies

$$p_S^{\boldsymbol{v}}(\boldsymbol{v}) = p_T^{\boldsymbol{v}}(\boldsymbol{v}), \quad p_S^{\widehat{\boldsymbol{c}}}(\widehat{\boldsymbol{c}}) = p_T^{\widehat{\boldsymbol{c}}}(\widehat{\boldsymbol{c}}),$$

or equivalently, $\boldsymbol{v} \perp u$ and $\widehat{\boldsymbol{c}} \perp u$.

Relaxed alignment ensures that CUDA is robust to concept differences across domains while maintaining alignment (more empirical results in Sec. 5).

**Full Game.** For any given $E$, we then derive the property of the optimal predictor $F$ and establish a tight lower bound for the prediction loss.

**Lemma 4.2** (**Optimal Predictor**). *Given the concept embedding encoder $E$, the prediction loss $\mathcal{L}_p(E, F)$ has a tight lower bound*

$$\mathcal{L}_p(E, F) \triangleq \mathbb{E}_S\left[L_p(F(E(\boldsymbol{x})), \boldsymbol{y})\right] \geq H(\boldsymbol{y} \mid E(\boldsymbol{x})),$$

*where $H(\cdot | \cdot)$ denotes the conditional entropy. The optimal predictor $F^*$ that minimizes the prediction loss is*

$$F^*(E(\boldsymbol{x})) = [\mathbb{P}(y_i = 1 \mid E(\boldsymbol{x}))]_{i=1}^Q,$$

*where $y_i$ denotes the $i$-th element of $\boldsymbol{y}$.*

Assuming the discriminator $D$ and the predictor $F$ are trained to achieve their optimum by Lemma 4.1 and Lemma 4.2, Eqn. 4 and Eqn. 5 can then be rewritten as:

$$\min_{E_{prob}} \mathcal{L}_c(E_{prob}), \tag{15}$$

$$\min_E H(\boldsymbol{y} \mid E(\boldsymbol{x})) - \lambda_d \cdot \widetilde{C}_d(E), \tag{16}$$

where $\widetilde{C}_d(E)$ is defined in Eqn. 12. With Eqn. 15~16 above, Theorem 4.2 below analyzes our optimal concept probability and embedding encoders $E$ and $E_{prob}$.

**Theorem 4.2** (**Optimal Concept Embedding Encoder**). *Assuming $u \perp \boldsymbol{y}$, if the concept embedding encoder $E$, concept probability encoder $E_{prob}$, the predictor $F$ and the discriminator $D$ have enough capacity and are trained to reach optimum, any global optimal concept embedding encoder $E^*$ and its corresponding global optimal concept probability encoder $E_{prob}^*$ have the following properties:*

$$E_{prob}^*(\boldsymbol{x}) = [\mathbb{P}(c_i = 1 | \boldsymbol{x})]_{i=1}^K, \tag{17}$$

$$H\left(\boldsymbol{y} \mid E^*(\boldsymbol{x})\right) = H(\boldsymbol{y} \mid \boldsymbol{x}), \tag{18}$$

$$\widetilde{C}_d\left(E^*\right) = \max_{E'} \widetilde{C}_d\left(E'\right). \tag{19}$$

Theorem 4.2 shows that, at equilibrium, (1) the optimal concept probability encoder $E_{prob}^*$ recovers the conditional distribution of the ground-truth concepts, and (2) the optimal concept embedding encoder $E^*$ preserves all the information about label $\boldsymbol{y}$ contained in the data $\boldsymbol{x}$.

*Table 1.* Performance of concept-based methods on both concept learning and classification across different datasets. CEM (w/o R.) indicates "without RandInt". I-II, III-IV and V-VI indicate different skin tone scale in the Fitzpatrick dataset. We mark the best result with **bold face** and the second best results with underline. Average accuracy is calculated over every three datasets of the same type images.

| Datasets | Waterbirds-2 | | | Waterbirds-200 | | | Waterbirds-CUB | | | AVG |
|---|---|---|---|---|---|---|---|---|---|---|
| Metrics | Concept | Concept F1 | Class | Concept | Concept F1 | Class | Concept | Concept F1 | Class | ACC |
| CEM | $94.14_{\pm0.13}$ | $81.74_{\pm0.39}$ | $70.27_{\pm1.70}$ | $93.68_{\pm0.10}$ | $81.22_{\pm0.64}$ | $62.26_{\pm1.11}$ | $93.64_{\pm0.08}$ | $80.08_{\pm0.34}$ | $66.48_{\pm0.81}$ | 66.34 |
| CEM (w/o R.) | $94.17_{\pm0.14}$ | $81.96_{\pm0.30}$ | $69.45_{\pm2.15}$ | $93.76_{\pm0.20}$ | $81.04_{\pm0.82}$ | $63.56_{\pm1.25}$ | $93.66_{\pm0.14}$ | $79.80_{\pm0.36}$ | $65.89_{\pm0.51}$ | 66.30 |
| CBM | $93.60_{\pm0.20}$ | $83.89_{\pm0.49}$ | $74.81_{\pm2.16}$ | $93.50_{\pm0.16}$ | $83.14_{\pm0.98}$ | $63.89_{\pm1.16}$ | $93.40_{\pm0.14}$ | $82.10_{\pm0.48}$ | $63.89_{\pm1.00}$ | 67.53 |
| **CUDA (Ours)** | $\mathbf{94.63_{\pm0.05}}$ | $\mathbf{84.97_{\pm0.15}}$ | $\mathbf{92.90_{\pm0.31}}$ | $\mathbf{95.15_{\pm0.05}}$ | $\mathbf{85.06_{\pm0.19}}$ | $\mathbf{75.87_{\pm0.31}}$ | $\mathbf{94.58_{\pm0.07}}$ | $\mathbf{82.81_{\pm0.19}}$ | $\mathbf{74.66_{\pm0.19}}$ | **81.15** |

| Datasets | MNIST → MNIST-M | | | SVHN → MNIST | | | MNIST → USPS | | | AVG |
|---|---|---|---|---|---|---|---|---|---|---|
| Metrics | Concept | Concept F1 | Class | Concept | Concept F1 | Class | Concept | Concept F1 | Class | ACC |
| CEM | $86.55_{\pm1.01}$ | $72.97_{\pm1.46}$ | $50.81_{\pm1.46}$ | $89.20_{\pm1.01}$ | $78.99_{\pm2.19}$ | $67.58_{\pm2.91}$ | $93.08_{\pm0.60}$ | $85.27_{\pm0.69}$ | $73.71_{\pm3.35}$ | 64.03 |
| CEM (w/o R.) | $86.40_{\pm1.01}$ | $72.58_{\pm1.01}$ | $49.36_{\pm2.39}$ | $89.89_{\pm2.20}$ | $80.22_{\pm4.31}$ | $69.76_{\pm5.30}$ | $92.65_{\pm1.98}$ | $83.75_{\pm3.83}$ | $72.92_{\pm8.65}$ | 64.01 |
| CBM | $86.28_{\pm0.22}$ | $72.86_{\pm0.22}$ | $49.66_{\pm2.18}$ | $89.63_{\pm0.93}$ | $79.51_{\pm1.70}$ | $65.03_{\pm2.94}$ | $90.67_{\pm2.78}$ | $79.34_{\pm6.35}$ | $61.79_{\pm14.24}$ | 58.82 |
| **CUDA (Ours)** | $\mathbf{98.51_{\pm0.02}}$ | $\mathbf{97.20_{\pm0.02}}$ | $\mathbf{95.24_{\pm0.13}}$ | $\mathbf{95.22_{\pm0.24}}$ | $\mathbf{90.95_{\pm0.24}}$ | $\mathbf{82.49_{\pm0.27}}$ | $\mathbf{98.78_{\pm0.03}}$ | $\mathbf{97.46_{\pm0.09}}$ | $\mathbf{96.01_{\pm0.13}}$ | **91.25** |

| Datasets | I-II → III-IV | | | III-IV → V-VI | | | III-IV → I-II | | | AVG |
|---|---|---|---|---|---|---|---|---|---|---|
| Metrics | Concept | Concept F1 | Class | Concept | Concept F1 | Class | Concept | Concept F1 | Class | ACC |
| CEM | $93.81_{\pm0.16}$ | $52.04_{\pm0.26}$ | $73.41_{\pm0.93}$ | $93.05_{\pm0.02}$ | $56.46_{\pm0.19}$ | $76.27_{\pm0.17}$ | $93.85_{\pm0.16}$ | $54.32_{\pm0.22}$ | $71.31_{\pm0.50}$ | 73.67 |
| CEM (w/o R.) | $93.78_{\pm0.17}$ | $51.98_{\pm0.27}$ | $73.13_{\pm0.63}$ | $93.05_{\pm0.02}$ | $56.47_{\pm0.15}$ | $76.86_{\pm1.19}$ | $93.80_{\pm0.13}$ | $54.26_{\pm0.18}$ | $71.72_{\pm0.38}$ | 73.91 |
| CBM | $94.11_{\pm0.43}$ | $52.17_{\pm0.68}$ | $72.37_{\pm0.00}$ | $92.27_{\pm0.57}$ | $56.21_{\pm0.57}$ | $78.82_{\pm0.00}$ | $94.16_{\pm0.34}$ | $54.27_{\pm0.20}$ | $70.49_{\pm0.00}$ | 73.89 |
| **CUDA (Ours)** | $\mathbf{95.37_{\pm0.07}}$ | $\mathbf{79.91_{\pm0.16}}$ | $\mathbf{78.85_{\pm0.31}}$ | $\mathbf{94.62_{\pm0.01}}$ | $\mathbf{79.57_{\pm0.25}}$ | $\mathbf{80.58_{\pm0.72}}$ | $\mathbf{95.45_{\pm0.06}}$ | $\mathbf{80.17_{\pm0.22}}$ | $\mathbf{76.53_{\pm0.49}}$ | **78.65** |

*Table 2.* Classification accuracy across different datasets. Zero-shot predictor is one of the baselines and components of CONDA. We mark the best result with **bold face** and the second best results with underline. Average accuracy is calculated over every three datasets of the same type images. Note that these baselines do not have concept accuracy and F1 because they cannot predict concepts directly.

| Model \ Dataset | WB-2 | WB-200 | WB-CUB | AVG | M → M-M | S → M | M → U | AVG | I-II → III-IV | III-IV → V-VI | III-IV → I-II | AVG |
|---|---|---|---|---|---|---|---|---|---|---|---|---|
| Zero-shot | $59.27_{\pm0.00}$ | $1.93_{\pm0.00}$ | $2.11_{\pm0.00}$ | 21.10 | $11.60_{\pm0.00}$ | $13.16_{\pm0.00}$ | $13.15_{\pm0.00}$ | 12.64 | $69.84_{\pm0.00}$ | $72.50_{\pm0.00}$ | $72.50_{\pm0.00}$ | 71.61 |
| PCBM | $53.08_{\pm1.89}$ | $28.99_{\pm0.53}$ | $34.60_{\pm0.45}$ | 38.89 | $29.66_{\pm1.02}$ | $21.32_{\pm2.12}$ | $15.55_{\pm0.12}$ | 22.18 | $72.13_{\pm0.33}$ | $72.64_{\pm0.14}$ | $72.64_{\pm0.14}$ | 72.47 |
| CONDA | $70.23_{\pm0.17}$ | $0.79_{\pm0.05}$ | $0.43_{\pm0.02}$ | 23.82 | $9.75_{\pm0.00}$ | $9.80_{\pm0.00}$ | $17.89_{\pm0.00}$ | 12.48 | $13.12_{\pm0.00}$ | $14.58_{\pm0.00}$ | $14.58_{\pm0.00}$ | 14.09 |
| DANN | $48.08_{\pm0.89}$ | $67.19_{\pm0.80}$ | $64.52_{\pm0.23}$ | 59.93 | $37.57_{\pm1.13}$ | $78.05_{\pm2.89}$ | $73.96_{\pm2.66}$ | 63.19 | $75.76_{\pm0.34}$ | $79.16_{\pm0.11}$ | $73.29_{\pm0.29}$ | 76.07 |
| MCD | $55.96_{\pm2.63}$ | $64.87_{\pm0.37}$ | $64.31_{\pm0.18}$ | 61.71 | $51.08_{\pm2.53}$ | $80.20_{\pm2.08}$ | $93.90_{\pm0.25}$ | 75.06 | $75.12_{\pm0.24}$ | $78.14_{\pm0.11}$ | $72.34_{\pm0.16}$ | 75.20 |
| SRDC | $48.49_{\pm0.54}$ | $73.29_{\pm0.73}$ | $69.42_{\pm0.77}$ | 63.73 | $30.35_{\pm0.88}$ | $78.99_{\pm0.72}$ | $93.71_{\pm0.54}$ | 67.68 | $73.70_{\pm0.29}$ | $78.69_{\pm0.40}$ | $72.91_{\pm0.16}$ | 75.10 |
| UTEP | $43.50_{\pm0.33}$ | $69.09_{\pm0.42}$ | $35.28_{\pm0.25}$ | 49.29 | $65.98_{\pm2.26}$ | $66.35_{\pm0.91}$ | $95.04_{\pm0.63}$ | 75.79 | $76.34_{\pm0.34}$ | $80.34_{\pm0.29}$ | $74.66_{\pm0.27}$ | 77.11 |
| GH++ | $45.65_{\pm1.13}$ | $79.87_{\pm0.35}$ | $79.46_{\pm0.43}$ | 68.33 | $59.40_{\pm0.86}$ | $79.12_{\pm0.86}$ | $93.35_{\pm0.59}$ | 77.29 | $75.98_{\pm0.57}$ | $78.76_{\pm0.69}$ | $75.04_{\pm0.68}$ | 76.59 |
| **CUDA (Ours)** | $92.90_{\pm0.31}$ | $75.87_{\pm0.31}$ | $74.66_{\pm0.19}$ | 81.15 | $95.24_{\pm0.13}$ | $82.49_{\pm0.27}$ | $96.01_{\pm0.13}$ | 91.25 | $78.85_{\pm0.31}$ | $80.58_{\pm0.72}$ | $76.53_{\pm0.49}$ | 78.65 |

# 5. Experiments

We evaluate CUDA across eight real-world datasets.

## 5.1. Evaluation Setup

**Datasets.** The original *Waterbirds* dataset (Sagawa et al., 2019) is split into a source domain and a target domain (Waterbirds-shift), by selecting images with opposite label and background; it only includes binary labels and does not have any concept information. To evaluate concept-based DA, we augment the *Waterbirds* dataset by incorporating concepts from the *CUB* dataset (Wah et al., 2011), leading to three datasets:

- **Waterbirds-2** is similar to the original *Waterbirds* with binary classification, i.e., landbirds/waterbirds,
- **Waterbirds-200** is the augmented version of *Waterbirds* with 200-class labels from CUB, and
- **Waterbirds-CUB** contains CUB training data as the source domain and Waterbirds-shift as the target.

We also use digit image datasets, including **MNIST** (Le-Cun et al., 1998), **MNIST-M** (Ganin et al., 2016), **SVHN**

(Netzer et al., 2011), and **USPS** (Hull, 1994), as different source and target domains. Since the target labels represent the digits 0-9, we design 11 topology concepts based on these datasets. Besides, we use **SkinCON** (Daneshjou et al., 2022b) to evaluate our approach in the medical domain. SkinCON includes 48 concepts selected by two dermatologists, annotated on the Fitzpatrick 17k dataset (Groh et al., 2021). For our experiments, we use one skin tone as the source domain and another as the target domain. Additional details are provided in Appendix C.1.

**Baselines and Implementation Details.** For concept-based baselines, we include **CBMs** (Koh et al., 2020), **CEMs** (Zarlenga et al., 2022), and **PCBMs** (Yuksekgonul et al., 2022). Additionally, we use state-of-the-art unsupervised domain adaption methods as baselines, including **DANN** (Ganin et al., 2016), **MCD** (Saito et al., 2018), **SRDC** (Tang et al., 2020), **UTEP** (Hu et al., 2022), and **GH++** (Huang et al., 2024). We also include **CONDA** (Choi et al., 2024), which performs test-time adaptation on PCBMs. Collectively, these methods define a comprehensive benchmark for domain adaptation in the context of concept learning. We summarize the implementation details in Appendix C.2.

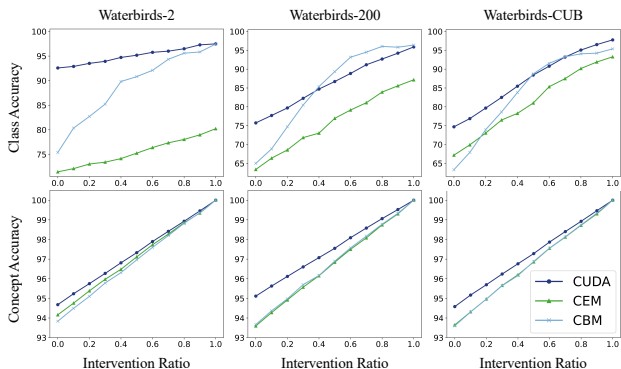

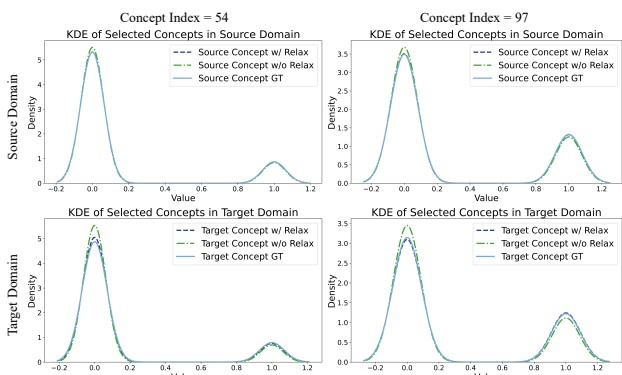

*Figure 3.* Concept intervention performance with different ratios of intervened concepts on Watebirds datasets. The intervention ratio denotes the proportion of provided correct concepts.

*Figure 4.* The kernel density estimation (KDE) plots compare the distributions of two selected concept indices under three different scenarios: Ground-truth (GT), without relaxation (w/o Relax), and with relaxation (w/ Relax).

**Evaluation Metrics.** We calculate concept accuracy and the related concept F1 score to assess the concept learning process. Note that only concept-based methods, i.e., CEM, CBM, and CUDA, have concept accuracy and concept F1. We also use class accuracy to evaluate the model's prediction accuracy. All metrics are computed on the *target domain*.

## 5.2. Results

**Prediction.** Tables 1 and 2 summarize the results. Table 1 shows that our CUDA performs exceptionally well within the CBM category, achieving state-of-the-art performance across all metrics. Notably, it outperforms other CBMs by a significant margin on the Waterbirds and MNIST datasets, while demonstrating consistent improvements on SkinCON. These results highlight the effectiveness of our method in learning concepts and adapting to domain shifts.

The upper section of Table 2 shows results for PCBM methods. Although PCBMs utilize concept banks to improve the efficiency of concept learning, their applicability to real-world domain adaptation tasks is limited, with performance falling short of standard CBMs. While CONDA incorporates test-time adaptation, its effectiveness is inconsistent, and its robustness is inferior to that of vanilla PCBMs. This underscores the importance of learning meaningful concept embeddings – merely compressing concepts does not work well for domain adaptation tasks.

The lower section of Table 2 shows results for DA methods and our concept-based CUDA. While DA models outperform some concept-based baselines, CUDA remains competitive, achieving the highest average accuracy across each type of the datasets. Note that existing DA methods cannot learn interpretable concepts, making them challenging to apply in high-risk scenarios. Our CUDA addresses this limitation, ensuring interpretability without compromising performance. Limitations and future works are discussed in Appendix D.

**Concept Intervention.** Concept intervention is a key task to evaluate concept-based interpretability, where users in-

tervene on (modify) specific predicted concepts to correct model predictions. Our CUDA is also capable of concept intervention while traditional DA is not. Similar to CBMs and CEMs (Koh et al., 2020; Zarlenga et al., 2022), we use ground-truth concepts with varying proportions at test-time to conduct interventions. Fig. 3 shows the performance of different methods after intervening on (correcting) varying proportions of concepts, referred to as intervention ratios. Our CUDA significantly outperforms the baselines across all intervention ratios in terms of both concept accuracy and classification accuracy.

**Alignment Relaxation.** In Theorem 4.1, we discussed the relaxation on the discriminator loss to account for concept differences. Fig. 4 illustrates the distributions of two selected concept indices under three scenarios: ground-truth (GT), without relaxation (w/o Relax), and with relaxation (w/ Relax). The GT distribution serves as a reference to evaluate the impact of relaxation on concept representations. The curves demonstrate how the relaxation process influences the density distribution of the concepts. Specifically, our relaxed alignment allows for greater differences between source and target concept distributions; such flexibility leads to predicted concept distributions closer to the ground truth and therefore higher final classification accuracy.

## 6. Conclusion

In this work, we proposed the **Concept-based Unsupervised Domain Adaptation (CUDA)** framework to address the challenges of generalization problem in Concept Bottleneck Models (CBMs). By aligning concept embeddings across domains through adversarial training and relaxing strict uniform alignment assumptions, CUDA enables CBMs to generalize effectively without requiring labeled concept data in the target domain. Our approach establishes new benchmarks for concept-based domain adaptation, significantly outperforming state-of-the-art CBM and DA methods while enhancing both interpretability and robustness.

## Impact Statement

This paper presents work whose goal is to advance the field of Machine Learning. There are many potential societal consequences of our work, none which we feel must be specifically highlighted here.

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

# A. Notation Table

*Table 3.* Main notations used in the method section. Click here to return to the main paper.

| Notation | Meaning |
|---|---|
| $\mathcal{X}$ | Input space |
| $\mathcal{Y}$ | Label space |
| $\mathcal{C}$ | Concept space |
| $\mathcal{V}$ | Concept embedding space |
| $\mathcal{H}$ | Hypothesis space |
| $n$ | Number of source domain data |
| $m$ | Number of target domain data |
| $K$ | Number of concepts |
| $Q$ | Number of classes |
| $J$ | Dimension of concept embedding |
| $\boldsymbol{c}$ | Ground-truth concepts |
| $\widehat{\boldsymbol{c}}$ | Concept predictions |
| $\boldsymbol{v}_i^{(+)}/\boldsymbol{v}_i^{(-)}$ | The positive/ negative concept embedding of the $i$-th concept $c_i$ |
| $\boldsymbol{v}$ | Concept embedding with predicted concepts |
| $\boldsymbol{v^c}$ | Concept embedding with ground-truth concepts |
| $E$ | Concept embedding encoder |
| $E_{prob}$ | Concept probability encoder |
| $F$ | Label predictor |
| $D$ | Domain discriminator |
| $\mathcal{D}_S/\mathcal{D}_T$ | Source/Target domain distribution over $\mathcal{X}$ |
| $f_S/f_T$ | Source/Target domain labeling function over $\mathcal{X}$ |
| $\widetilde{\mathcal{D}}_S/\widetilde{\mathcal{D}}_T$ | Source/Target domain distribution over $\mathcal{V}$ |
| $\widetilde{f}_S/\widetilde{f}_T$ | Source/Target domain labeling function over $\mathcal{V}$ |
| $\widetilde{\mathcal{D}}_S^{\boldsymbol{c}}$ | Source domain distribution over $\mathcal{V}$ with ground-truth concepts |
| $\widetilde{f}_S^{\boldsymbol{c}}$ | Source domain labeling function over $\mathcal{V}$ with ground-truth concepts |
| $h$ | Hypothesis function |
| $\epsilon_S$ | Source error |
| $\epsilon_T$ | Target error |
| $\epsilon_S^{\boldsymbol{c}}$ | Source error with ground-truth concepts |

# B. Proof

## B.1. Proof of Generalization Error Bound for CBMs

**Lemma 3.1** (**Source Error with Predicted Concept Embeddings**). *Let $\mathcal{H}$ be a hypothesis space where all hypotheses $h \in \mathcal{H}$ are $L$-Lipschitz continuous under the Euclidean norm $\|\cdot\|_2$ for some constant $L > 0$. Assume that for all $\boldsymbol{v} \in \mathcal{V}$, $\|\boldsymbol{v}\|_2$ is bounded. Then, for any $h_1, h_2 \in \mathcal{H}$, there exists a finite constant $r > 0$ such that*

$$\epsilon_S(h_1, h_2) \leq \epsilon_S^{\boldsymbol{c}}(h_1, h_2) + r \cdot \mathbb{E}_S\left[\|\widehat{\boldsymbol{c}} - \boldsymbol{c}\|_2\right],$$

*where $\epsilon_S(h_1, h_2) = \mathbb{E}_{\boldsymbol{v} \sim \widetilde{\mathcal{D}}_S}\left[|h_1(\boldsymbol{v}) - h_2(\boldsymbol{v})|\right]$ and $\epsilon_S^{\boldsymbol{c}}(h_1, h_2) = \mathbb{E}_{\boldsymbol{v^c} \sim \widetilde{\mathcal{D}}_S^{\boldsymbol{c}}}\left[|h_1(\boldsymbol{v^c}) - h_2(\boldsymbol{v^c})|\right]$ are the disagreement between hypotheses $h_1$ and $h_2$ w.r.t. distributions $\widetilde{D}_S$ and $\widetilde{D}_S^{\boldsymbol{c}}$, respectively, and $\mathbb{E}_S$ denotes the expectation taken over the source distribution.*

*Proof.* Note that the concept embedding with the ground-truth concepts $\boldsymbol{v^c}$ and the concept embedding with the predicted concepts $\boldsymbol{v}$ are defined as follows:

$$\boldsymbol{v^c} = \left[\left(c_1\boldsymbol{v}_1^{(+)} + (1-c_1)\boldsymbol{v}_1^{(-)}\right)^{\mathsf{T}}, \ldots, \left(c_K\boldsymbol{v}_K^{(+)} + (1-c_K)\boldsymbol{v}_K^{(-)}\right)^{\mathsf{T}}\right]^{\mathsf{T}},$$

$$\boldsymbol{v} = \left[\left(\widehat{c}_1\boldsymbol{v}_1^{(+)} + (1-\widehat{c}_1)\boldsymbol{v}_1^{(-)}\right)^{\mathsf{T}}, \ldots, \left(\widehat{c}_K\boldsymbol{v}_K^{(+)} + (1-\widehat{c}_K)\boldsymbol{v}_K^{(-)}\right)^{\mathsf{T}}\right]^{\mathsf{T}},$$

where $v$ and $v^c$ share the same $v^{(+)} = \left[ v_1^{(+)\mathsf{T}}, \ldots, v_K^{(+)\mathsf{T}} \right]^{\mathsf{T}}$ and $v^{(-)} = \left[ v_1^{(-)\mathsf{T}}, \ldots, v_K^{(-)\mathsf{T}} \right]^{\mathsf{T}}$. Then, $\epsilon_S(h_1, h_2)$ with respect to arbitrary concept embedding $v$ can be upper bounded by

$$
\begin{aligned}
\epsilon_S(h_1, h_2) &= \mathbb{E}_{v \sim \widetilde{\mathcal{D}}_S} \left[ |h_1(v) - h_2(v)| \right] \\
&= \mathbb{E}_{v \sim \widetilde{\mathcal{D}}_S, v^c \sim \widetilde{\mathcal{D}}_S^c} \left[ |h_1(v) - h_1(v^c) + h_1(v^c) - h_2(v^c) + h_2(v^c) - h_2(v)| \right] \\
&\leq \mathbb{E}_{v \sim \widetilde{\mathcal{D}}_S, v^c \sim \widetilde{\mathcal{D}}_S^c} \left[ |h_1(v) - h_1(v^c)| + |h_1(v^c) - h_2(v^c)| + |h_2(v^c) - h_2(v)| \right] \\
&\overset{(i)}{\leq} 2L \cdot \mathbb{E}_{v \sim \widetilde{\mathcal{D}}_S, v^c \sim \widetilde{\mathcal{D}}_S^c} \left[ \|v^c - v\|_2 \right] + \mathbb{E}_{v^c \sim \widetilde{\mathcal{D}}_S^c} \left[ |h_1(v^c) - h_2(v^c)| \right] \\
&= 2L \cdot \mathbb{E}_{v \sim \widetilde{\mathcal{D}}_S, v^c \sim \widetilde{\mathcal{D}}_S^c} \left[ \|v^c - v\|_2 \right] + \epsilon_S^c(h_1, h_2),
\end{aligned}
\tag{20}
$$

where $(i)$ is due to the Lipschitz continuity of $h_1, h_2 \in \mathcal{H}$ with a constant $L > 0$, and $\epsilon_S^c(h_1, h_2) \triangleq \mathbb{E}_{v^c \sim \widetilde{\mathcal{D}}_S^c} \left[ |h_1(v^c) - h_2(v^c)| \right]$. Note that for the $i$-th concept, $v_i = \widehat{c}_i v_i^{(+)} + (1 - \widehat{c}_i) v_i^{(-)}$ and $v_i^c = c_i v_i^{(+)} + (1 - c_i) v_i^{(-)}$. Thus, $v_i - v_i^c = (\widehat{c}_i - c_i) \left( v_i^{(+)} - v_i^{(-)} \right)$. Because we assume for all $v = [v_i]_{i=1}^K \in \mathcal{V}$, $\|v\|_2$ is bounded. There exists a sufficiently large $M$, such that $\max_i \left\| v_i^{(+)} - v_i^{(-)} \right\|_2 \leq M$. Then, the difference between the concept embedding with the ground-truth concepts and that with the predicted concepts under the Euclidean norm has the following upper bound:

$$
\begin{aligned}
\|v - v^c\|_2 &= \left\| \left[ (v_1 - v_1^c)^{\mathsf{T}}, \ldots, (v_K - v_K^c)^{\mathsf{T}} \right]^{\mathsf{T}} \right\|_2 \\
&= \left\| \left[ (\widehat{c}_1 - c_1) \cdot \left( v_1^{(+)} - v_1^{(-)} \right)^{\mathsf{T}}, \ldots, (\widehat{c}_K - c_K) \cdot \left( v_K^{(+)} - v_K^{(-)} \right)^{\mathsf{T}} \right]^{\mathsf{T}} \right\|_2 \\
&\leq M \cdot \|\widehat{c} - c\|_2.
\end{aligned}
\tag{21}
$$

Plugging Eqn. 21 into Eqn. 20 and then we can get

$$
\begin{aligned}
\epsilon_S(h_1, h_2) &\leq 2L \cdot \mathbb{E}_{v \sim \widetilde{\mathcal{D}}_S, v^c \sim \widetilde{\mathcal{D}}_S^c} \left[ \|v^c - v\|_2 \right] + \epsilon_S^c(h_1, h_2) \\
&\leq 2LM \cdot \mathbb{E}_S \left[ \|\widehat{c} - c\|_2 \right] + \epsilon_S^c(h_1, h_2),
\end{aligned}
$$

where $c$ is only available in the source domain. Letting $r = 2LM$, we complete the proof. $\qquad \square$

**Theorem 3.1** (**Target-Domain Error Bound for Concept-Based Models**). *Under the assumption of Lemma 3.1, for any $h \in \mathcal{H}$, we have:*

$$
\begin{aligned}
\epsilon_T(h) &\leq \epsilon_S^c(h) + \tfrac{1}{2} d_{\mathcal{H}\Delta\mathcal{H}} \left( \widetilde{\mathcal{D}}_S^c, \widetilde{\mathcal{D}}_T \right) + \eta^c \\
&\quad + R \cdot \mathbb{E}_S \left[ \|\widehat{c} - c\|_2 \right],
\end{aligned}
\tag{3}
$$

*where $R > 0$ is a finite constant, $\eta^c = \min_{h \in \mathcal{H}} \epsilon_S^c(h) + \epsilon_T(h)$, and $d_{\mathcal{H}\triangle\mathcal{H}}(\widetilde{\mathcal{D}}_S^c, \widetilde{\mathcal{D}}_T)$ denotes the $\mathcal{H}\Delta\mathcal{H}$ divergence between distribution $\widetilde{\mathcal{D}}_S^c$ and distribution $\widetilde{\mathcal{D}}_T$.*

*Proof.* Let $h^* = \arg\min_{h \in \mathcal{H}} \epsilon_S^c(h) + \epsilon_T(h)$ and $\eta^c = \min_{h \in \mathcal{H}} \epsilon_S^c(h) + \epsilon_T(h) = \epsilon_S^c(h^*) + \epsilon_T(h^*)$. By the triangle inequality for classification error, i.e. $\epsilon(h_1, h_2) \leq \epsilon(h_1, h_3) + \epsilon(h_2, h_3)$, we have

$$
\begin{aligned}
\epsilon_T(h) &\leq \epsilon_T(h^*) + \epsilon_T(h, h^*) \\
&\leq \epsilon_T(h^*) + \epsilon_S(h, h^*) + |\epsilon_S(h, h^*) - \epsilon_T(h, h^*)|.
\end{aligned}
\tag{22}
$$

We define the source error for concept embedding constructed using ground-truth concepts as:

$$
\epsilon_S^c(h) \triangleq \epsilon_S^c(h, \widetilde{f}_S^c) = \mathbb{E}_{v^c \sim \widetilde{\mathcal{D}}_S^c} \left[ \left| \widetilde{f}_S^c(v^c) - h(v^c) \right| \right],
$$

where $\widetilde{\mathcal{D}}_S^c$ is the marginal distribution over $\boldsymbol{v}^c$ and $\widetilde{f}_S^c(\boldsymbol{v}^c) \triangleq \mathbb{E}_{\boldsymbol{x} \sim \mathcal{D}_S}[f_S(\boldsymbol{x}) \mid E(\boldsymbol{x}) = \boldsymbol{v}^c]$ is the corresponding induced labeling function. Note that $\widetilde{f}_S^c$ can also be a hypothesis. Then for the second term $\epsilon_S(h, h^*)$, we can bound it by the source error with ground-truth concepts:

$$
\begin{aligned}
\epsilon_S(h, h^*) &\leq \epsilon_S\left(h, \widetilde{f}_S^c\right) + \epsilon_S\left(h^*, \widetilde{f}_S^c\right) \\
&\leq \left(\left|\epsilon_S\left(h, \widetilde{f}_S^c\right) - \epsilon_S^c\left(h, \widetilde{f}_S^c\right)\right| + \epsilon_S^c\left(h, \widetilde{f}_S^c\right)\right) + \left(\left|\epsilon_S\left(h^*, \widetilde{f}_S^c\right) - \epsilon_S^c\left(h^*, \widetilde{f}_S^c\right)\right| + \epsilon_S^c\left(h^*, \widetilde{f}_S^c\right)\right) \quad (23) \\
&\overset{(i)}{\leq} \left(r_1 \cdot \mathbb{E}_S\left[\|\widehat{\boldsymbol{c}} - \boldsymbol{c}\|_2\right] + \epsilon_S^c(h)\right) + \left(r_2 \cdot \mathbb{E}_S\left[\|\widehat{\boldsymbol{c}} - \boldsymbol{c}\|_2\right] + \epsilon_S^c(h^*)\right),
\end{aligned}
$$

where $\epsilon_S^c\left(h, \widetilde{f}_S^c\right) = \epsilon_S^c(h)$ and $\epsilon_S^c\left(h^*, \widetilde{f}_S^c\right) = \epsilon_S^c(h^*)$, and (i) is due to Lemma 3.1: there exists finite constant $r_1, r_2$ such that $\left|\epsilon_S\left(h, \widetilde{f}_S^c\right) - \epsilon_S^c\left(h, \widetilde{f}_S^c\right)\right| \leq r_1 \cdot \mathbb{E}_S\left[\|\widehat{\boldsymbol{c}} - \boldsymbol{c}\|_2\right]$ and $\left|\epsilon_S\left(h^*, \widetilde{f}_S^c\right) - \epsilon_S^c\left(h^*, \widetilde{f}_S^c\right)\right| \leq r_2 \cdot \mathbb{E}_S\left[\|\widehat{\boldsymbol{c}} - \boldsymbol{c}\|_2\right]$. By the definition of $\mathcal{H}\Delta\mathcal{H}$ divergence (Ben-David et al., 2010):

$$
d_{\mathcal{H}\Delta\mathcal{H}}\left(\widetilde{\mathcal{D}}_S^c, \widetilde{\mathcal{D}}_T\right) \triangleq 2 \sup_{h, h' \in \mathcal{H}} \left|\mathbb{P}_{\boldsymbol{v} \sim \widetilde{\mathcal{D}}_T}[h(\boldsymbol{v}) \neq h'(\boldsymbol{v})] - \mathbb{P}_{\boldsymbol{v}^c \sim \widetilde{\mathcal{D}}_S^c}[h(\boldsymbol{v}^c) \neq h'(\boldsymbol{v}^c)]\right|,
$$

the last term of Eqn. 22 is bounded by

$$
\begin{aligned}
|\epsilon_S(h, h^*) - \epsilon_T(h, h^*)| &\leq |\epsilon_S(h, h^*) - \epsilon_S^c(h, h^*)| + |\epsilon_T(h, h^*) - \epsilon_S^c(h, h^*)| \\
&\overset{(ii)}{\leq} r_3 \cdot \mathbb{E}_S\left[\|\widehat{\boldsymbol{c}} - \boldsymbol{c}\|_2\right] + |\epsilon_T(h, h^*) - \epsilon_S^c(h, h^*)| \\
&\leq r_3 \cdot \mathbb{E}_S\left[\|\widehat{\boldsymbol{c}} - \boldsymbol{c}\|_2\right] + \sup_{h, h' \in \mathcal{H}} |\epsilon_T(h, h') - \epsilon_S^c(h, h')| \\
&\leq r_3 \cdot \mathbb{E}_S\left[\|\widehat{\boldsymbol{c}} - \boldsymbol{c}\|_2\right] + \sup_{h, h' \in \mathcal{H}} \left|\mathbb{P}_{\boldsymbol{v} \sim \widetilde{\mathcal{D}}_T}[h(\boldsymbol{v}) \neq h'(\boldsymbol{v})] - \mathbb{P}_{\boldsymbol{v}^c \sim \widetilde{\mathcal{D}}_S^c}[h(\boldsymbol{v}^c) \neq h'(\boldsymbol{v}^c)]\right| \quad (24) \\
&= r_3 \cdot \mathbb{E}_S\left[\|\widehat{\boldsymbol{c}} - \boldsymbol{c}\|_2\right] + \tfrac{1}{2} d_{\mathcal{H}\Delta\mathcal{H}}\left(\widetilde{\mathcal{D}}_S^c, \widetilde{\mathcal{D}}_T\right).
\end{aligned}
$$

where (ii) is also due to Lemma 3.1 with the constant $r = r_3$. Plugging Eqn. 23 and Eqn. 24 into Eqn. 22, then we can obtain the final upper bound of target error for CBMs:

$$
\begin{aligned}
\epsilon_T(h) &\leq \epsilon_T(h^*) + \epsilon_S(h, h^*) + |\epsilon_S(h, h^*) - \epsilon_T(h, h^*)| \\
&\leq \epsilon_T(h^*) + \left(r_1 \cdot \mathbb{E}_S\left[\|\widehat{\boldsymbol{c}} - \boldsymbol{c}\|_2\right] + \epsilon_S^c(h)\right) + \left(r_2 \cdot \mathbb{E}_S\left[\|\widehat{\boldsymbol{c}} - \boldsymbol{c}\|_2\right] + \epsilon_S^c(h^*)\right) + r_3 \cdot \mathbb{E}_S\left[\|\widehat{\boldsymbol{c}} - \boldsymbol{c}\|_2\right] + \tfrac{1}{2} d_{\mathcal{H}\Delta\mathcal{H}}\left(\widetilde{\mathcal{D}}_S^c, \widetilde{\mathcal{D}}_T\right) \\
&= \epsilon_S^c(h) + \epsilon_S^c(h^*) + \epsilon_T(h^*) + R \cdot \mathbb{E}_S\left[\|\widehat{\boldsymbol{c}} - \boldsymbol{c}\|_2\right] + \tfrac{1}{2} d_{\mathcal{H}\Delta\mathcal{H}}\left(\widetilde{\mathcal{D}}_S^c, \widetilde{\mathcal{D}}_T\right) \\
&= \epsilon_S^c(h) + \eta^c + R \cdot \mathbb{E}_S\left[\|\widehat{\boldsymbol{c}} - \boldsymbol{c}\|_2\right] + \tfrac{1}{2} d_{\mathcal{H}\Delta\mathcal{H}}\left(\widetilde{\mathcal{D}}_S^c, \widetilde{\mathcal{D}}_T\right),
\end{aligned}
$$

where $R = r_1 + r_2 + r_3$ and $\eta^c = \epsilon_S^c(h^*) + \epsilon_T(h^*)$, completing the proof. $\qquad\square$

## B.2. Proof of Theoretical Analysis for CUDA

**Lemma 4.1** (**Optimal Discriminator**). *For E fixed, the optimal discriminator D is*

$$
D_E^*(\boldsymbol{v}) = \frac{p_T^{\boldsymbol{v}}(\boldsymbol{v})}{p_S^{\boldsymbol{v}}(\boldsymbol{v}) + p_T^{\boldsymbol{v}}(\boldsymbol{v})},
$$

*where $p_S^{\boldsymbol{v}}(\boldsymbol{v})$ and $p_T^{\boldsymbol{v}}(\boldsymbol{v})$ are the probability density function of $\boldsymbol{v}$ in source and target domains, respectively.*

*Proof.* With $E$ fixed, the optimal $D$ should be

$$
\begin{aligned}
D_E^* &= \arg\min_D \mathbb{E}_{(\boldsymbol{x},u)\sim p(\boldsymbol{x},u)} \left[ L_d(D(E(\boldsymbol{x})), u) \right] \\
&= \arg\min_D \mathbb{E}_{(\boldsymbol{x},u)\sim p(\boldsymbol{x},u)} \left[ u \log \frac{1}{D(E(\boldsymbol{x}))} + (1-u) \log \frac{1}{1-D(E(\boldsymbol{x}))} \right] \\
&= \arg\min_D \mathbb{E}_{\boldsymbol{v}\sim p(\boldsymbol{v})} \left[ \mathbb{E}_{u\sim p(u|\boldsymbol{v})} \left[ u \log \frac{1}{D(\boldsymbol{v})} + (1-u) \log \frac{1}{1-D(\boldsymbol{v})} \right] \right] \\
&= \arg\min_D \mathbb{E}_{\boldsymbol{v}\sim p(\boldsymbol{v})} \left[ \mathbb{E}\left[u|\boldsymbol{v}\right] \cdot \log \frac{1}{D(\boldsymbol{v})} + (1-\mathbb{E}\left[u|\boldsymbol{v}\right]) \cdot \log \frac{1}{1-D(\boldsymbol{v})} \right] \\
&= \arg\max_D \mathbb{E}_{\boldsymbol{v}\sim p(\boldsymbol{v})} \left[ \mathbb{E}\left[u|\boldsymbol{v}\right] \cdot \log D(\boldsymbol{v}) + (1-\mathbb{E}\left[u|\boldsymbol{v}\right]) \cdot \log (1-D(\boldsymbol{v})) \right],
\end{aligned}
$$

where $\boldsymbol{v} = E(\boldsymbol{x})$. Note that for any $(a,b) \in \mathbb{R}^2 \backslash (0,0)$, the function $y \to a\log(1-y) + b\log(y)$ achieves its maximum in $[0,1]$ at $\frac{b}{a+b}$. Note that $\mathbb{P}(u=0) = \mathbb{P}(u=1) = \frac{1}{2}$, thus we have

$$
\begin{aligned}
D_E^*(\boldsymbol{v}) &= \mathbb{E}\left[u|\boldsymbol{v}\right] = \mathbb{P}\left(u=1|\boldsymbol{v}\right) \\
&\overset{(i)}{=} \frac{p(\boldsymbol{v}|u=1)\mathbb{P}(u=1)}{p(\boldsymbol{v}|u=1)\mathbb{P}(u=1)+p(\boldsymbol{v}|u=0)\mathbb{P}(u=0)} \\
&= \frac{p(\boldsymbol{v}|u=1)}{p(\boldsymbol{v}|u=1)+p(\boldsymbol{v}|u=0)} \\
&= \frac{p_T^{\boldsymbol{v}}(\boldsymbol{v})}{p_S^{\boldsymbol{v}}(\boldsymbol{v})+p_T^{\boldsymbol{v}}(\boldsymbol{v})},
\end{aligned}
$$

where $(i)$ is due to the Bayes rule, and the discriminator does not need to be defined outside of $Supp(p_S^{\boldsymbol{v}}(\boldsymbol{v})) \cup Supp(p_T^{\boldsymbol{v}}(\boldsymbol{v}))$.

$\square$

**Theorem 4.1** (**Relaxed Alignment**). *If the discriminator $D$ have enough capacity to be trained to reach optimum, the relaxed optimization objective $\widetilde{C}_d(E)$ defined in Eqn. 12 achieves its global maximum if and only if the concept embedding encoder satisfies the following conditions:*

$$
\text{JSD}(p_S^{\boldsymbol{v}}(\boldsymbol{v}) \| p_T^{\boldsymbol{v}}(\boldsymbol{v})) = \log 2 - \tau, \tag{13}
$$
$$
\text{JSD}(p_S^{\widehat{\boldsymbol{c}}}(\widehat{\boldsymbol{c}}) \| p_T^{\widehat{\boldsymbol{c}}}(\widehat{\boldsymbol{c}})) = \log 2 - \tau - I(\boldsymbol{v}, u|\widehat{\boldsymbol{c}}), \tag{14}
$$

*where $I(\cdot, \cdot|\cdot)$ is the conditional mutual information, $p_S^{\widehat{\boldsymbol{c}}}(\widehat{\boldsymbol{c}})$ and $p_T^{\widehat{\boldsymbol{c}}}(\widehat{\boldsymbol{c}})$ are the probability density function of $\widehat{\boldsymbol{c}}$ in source and target domains, respectively.*

*Proof.* If $D$ always achieves its optimum w.r.t $E$ during the training, we have

$$
\begin{aligned}
C_d(E) &\triangleq \min_D \mathcal{L}_d(E, D) = \mathcal{L}_d\left(E, D_E^*\right) \\
&= \mathbb{E}\left[L_d(D_E^*(E(\boldsymbol{x})), u)\right] \\
&= \mathbb{E}_{(\boldsymbol{v},u)\sim p(\boldsymbol{v},u)} \left[ u \log \frac{1}{D_E^*(\boldsymbol{v})} + (1-u) \log \frac{1}{1-D_E^*(\boldsymbol{v})} \right] \\
&= \mathbb{E}_{\boldsymbol{v}\sim p(\boldsymbol{v})} \left[ \mathbb{E}_{u\sim p(u|\boldsymbol{v})}[u] \cdot \log \frac{1}{\mathbb{E}_{u\sim p(u|\boldsymbol{v})}[u]} + \left(1 - \mathbb{E}_{u\sim p(u|\boldsymbol{v})}[u]\right) \cdot \log \frac{1}{1-\mathbb{E}_{u\sim p(u|\boldsymbol{v})}[u]} \right] \\
&= \mathbb{E}_{\boldsymbol{v}\sim p(\boldsymbol{v})} \left[ \mathbb{P}\left(u=1|\boldsymbol{v}\right) \cdot \log \frac{1}{\mathbb{P}(u=1|\boldsymbol{v})} + \mathbb{P}\left(u=0|\boldsymbol{v}\right) \cdot \log \frac{1}{\mathbb{P}(u=0|\boldsymbol{v})} \right] \\
&= H(u|\boldsymbol{v}) = H(u) - I(\boldsymbol{v}, u).
\end{aligned}
\tag{25}
$$

Note that $\mathbb{P}(u=1) = \mathbb{P}(u=0) = \frac{1}{2}$, then we have

$$
H(u) = \mathbb{P}(u=1) \cdot \log \frac{1}{\mathbb{P}(u=1)} + \mathbb{P}(u=0) \cdot \log \frac{1}{\mathbb{P}(u=0)} = \log 2,
$$

and

$$I(\boldsymbol{v}, u) \triangleq \mathbb{E}_{(\boldsymbol{v}, u)} \left[ \log \frac{p(u, \boldsymbol{v})}{p(u) \cdot p(\boldsymbol{v})} \right]$$

$$= \mathbb{E}_{u \sim p(u)} \left[ \mathbb{E}_{\boldsymbol{v} \sim p(\boldsymbol{v}|u)} \left[ \log \frac{p(\boldsymbol{v}|u)}{p(\boldsymbol{v})} \right] \right]$$

$$= \mathbb{E}_{u \sim p(u)} \left[ \mathrm{KL}(p(\boldsymbol{v} \mid u) \| p(\boldsymbol{v})) \right]$$

$$= \mathrm{KL}(p(\boldsymbol{v} \mid u = 1) \| p(\boldsymbol{v})) \cdot \mathbb{P}(u = 1) + \mathrm{KL}(p(\boldsymbol{v} \mid u = 0) \| p(\boldsymbol{v})) \cdot \mathbb{P}(u = 0)$$

$$= \frac{1}{2} \left( \mathrm{KL} \left( p(\boldsymbol{v} \mid u = 1) \| \frac{p(\boldsymbol{v}|u=1) + p(\boldsymbol{v}|u=0)}{2} \right) + \mathrm{KL} \left( p(\boldsymbol{v} \mid u = 0) \| \frac{p(\boldsymbol{v}|u=1) + p(\boldsymbol{v}|u=0)}{2} \right) \right)$$

$$= \mathrm{JSD} \left( p(\boldsymbol{v}|u = 1) \| p(\boldsymbol{v}|u = 0) \right)$$

$$= \mathrm{JSD} \left( p_T^{\boldsymbol{v}}(\boldsymbol{v}) \| p_S^{\boldsymbol{v}}(\boldsymbol{v}) \right),$$

where $p(\boldsymbol{v}) = p(\boldsymbol{v}|u = 1) \cdot \mathbb{P}(u = 1) + p(\boldsymbol{v}|u = 0) \cdot \mathbb{P}(u = 0) = \frac{p(\boldsymbol{v}|u=1) + p(\boldsymbol{v}|u=0)}{2}$, and JSD is short for Jensen–Shannon divergence, which is both non-negative and zero if and only if the two distributions are equal. Then Eqn. 25 can be rewritten as

$$C_d(E) = \log 2 - \mathrm{JSD} \left( p_T^{\boldsymbol{v}}(\boldsymbol{v}) \| p_S^{\boldsymbol{v}}(\boldsymbol{v}) \right).$$

To obtain the maximum of $C_d(E)$, $E$ should satisfy

$$p_S^{\boldsymbol{v}}(\boldsymbol{v}) = p_T^{\boldsymbol{v}}(\boldsymbol{v}),$$

and the corresponding maximum value equals $\log 2$. Thus, the relaxed objective $\widetilde{C}_d(E)$ defined in Eqn. 12:

$$\widetilde{C}_d(E) \triangleq \widetilde{\mathcal{L}}_d(E, D_E^*) = \min\{\mathcal{L}_d(E, D_E^*), \tau\} = \min\{C_d(E), \tau\}$$
$$= \min\{\log 2 - \mathrm{JSD} \left( p_T^{\boldsymbol{v}}(\boldsymbol{v}) \| p_S^{\boldsymbol{v}}(\boldsymbol{v}) \right), \tau\}$$

achieves its global maximum if and only if the concept embedding encoder satisfies:

$$\mathrm{JSD} \left( p_T^{\boldsymbol{v}}(\boldsymbol{v}) \| p_S^{\boldsymbol{v}}(\boldsymbol{v}) \right) = \log 2 - \tau. \tag{26}$$

Similarly, we can also obtain $I(\widehat{c}, u) = \mathrm{JSD} \left( p_T^{\widehat{c}}(\widehat{c}) \| p_S^{\widehat{c}}(\widehat{c}) \right)$. For the $i$-th concept, $\boldsymbol{v}_i^{(+)}$ and $\boldsymbol{v}_i^{(-)}$ are first mapped to $\widehat{c}_i$, which is then used to combine them into $\boldsymbol{v}_i$ as follows:

$$\boldsymbol{v}_i = \widehat{c}_i \boldsymbol{v}_i^{(+)} + (1 - \widehat{c}_i) \boldsymbol{v}_i^{(-)}.$$

This indicates that $\boldsymbol{v}$ contains all the information of $\widehat{c}$, and $H(u|\boldsymbol{v}) = H(u|\boldsymbol{v}, \widehat{c})$. Thus, we have

$$I(\boldsymbol{v}, u) = H(u) - H(u|\boldsymbol{v})$$
$$= H(u) - H(u|\boldsymbol{v}, \widehat{c})$$
$$= H(u) - H(u|\widehat{c}) + H(u|\widehat{c}) - H(u|\boldsymbol{v}, \widehat{c})$$
$$= I(\widehat{c}, u) + I(\boldsymbol{v}, u|\widehat{c}),$$

which is equivalent to

$$\mathrm{JSD} \left( p_T^{\boldsymbol{v}}(\boldsymbol{v}) \| p_S^{\boldsymbol{v}}(\boldsymbol{v}) \right) = \mathrm{JSD}(p_T^{\widehat{c}}(\widehat{c}) \| p_S^{\widehat{c}}(\widehat{c})) + I(\boldsymbol{v}, u|\widehat{c}). \tag{27}$$

Plugging Eqn. 26 into Eqn. 27, we finally obtain

$$\mathrm{JSD}(p_T^{\widehat{c}}(\widehat{c}) \| p_S^{\widehat{c}}(\widehat{c})) = \log 2 - \tau - I(\boldsymbol{v}, u|\widehat{c}),$$

completing the proof. $\qquad\square$

As for the special case for the theorem above, $\tau = \log 2$, it follows that

$$\mathrm{JSD} \left( p_T^{\boldsymbol{v}}(\boldsymbol{v}) \| p_S^{\boldsymbol{v}}(\boldsymbol{v}) \right) = 0,$$

which implies $\boldsymbol{v} \perp u$. Thus, in Eqn. 27 the last term $I(\boldsymbol{v}, u|\widehat{c}) = 0$, and

$$\mathrm{JSD} \left( p_T^{\widehat{c}}(\widehat{c}) \| p_S^{\widehat{c}}(\widehat{c}) \right) = \log 2 - \tau - I(\boldsymbol{v}, u|\widehat{c}) = 0 - 0 = 0,$$

which is equivalent to $\widehat{c} \perp u$.

**Lemma 4.2** (**Optimal Predictor**). *Given the concept embedding encoder $E$, the prediction loss $\mathcal{L}_p(E, F)$ has a tight lower bound*

$$\mathcal{L}_p(E, F) \triangleq \mathbb{E}_S\left[L_p(F(E(\boldsymbol{x})), \boldsymbol{y})\right] \geq H(\boldsymbol{y} \mid E(\boldsymbol{x})),$$

*where $H(\cdot|\cdot)$ denotes the conditional entropy. The optimal predictor $F^*$ that minimizes the prediction loss is*

$$F^*(E(\boldsymbol{x})) = [\mathbb{P}\left(y_i = 1 \mid E(\boldsymbol{x})\right)]_{i=1}^Q,$$

*where $y_i$ denotes the $i$-th element of $\boldsymbol{y}$.*

*Proof.* With $E$ fixed, the prediction loss $\mathcal{L}_p(E, F)$ can be rewritten as

$$\begin{aligned}
\mathcal{L}_p(E, F) &= \mathbb{E}\left[L_p(F(E(\boldsymbol{x})), \boldsymbol{y})\right] \\
&= \mathbb{E}_{(\boldsymbol{x}, \boldsymbol{y}) \sim p(\boldsymbol{x}, \boldsymbol{y})}\left[\sum_{i=1}^Q y_i \log \tfrac{1}{(F(E(\boldsymbol{x})))_i}\right] \\
&= \mathbb{E}_{(\boldsymbol{v}, \boldsymbol{y}) \sim p(\boldsymbol{v}, \boldsymbol{y})}\left[\sum_{i=1}^Q y_i \log \tfrac{1}{(F(\boldsymbol{v}))_i}\right] \\
&= \mathbb{E}_{\boldsymbol{v} \sim p(\boldsymbol{v})}\left[\sum_{i=1}^Q \mathbb{E}_{y_i \sim \mathbb{P}(y_i|\boldsymbol{v})}\left[y_i \log \tfrac{1}{(F(\boldsymbol{v}))_i}\right]\right] \\
&= \mathbb{E}_{\boldsymbol{v} \sim p(\boldsymbol{v})}\left[\sum_{i=1}^Q \mathbb{E}\left[y_i|\boldsymbol{v}\right] \log \tfrac{1}{(F(\boldsymbol{v}))_i}\right],
\end{aligned}$$

where $F(E(\boldsymbol{x})) = F(\boldsymbol{v}) \in \mathbb{R}^Q$, and we denote the $i$-th component of $F(\boldsymbol{v})$ as $(F(\boldsymbol{v}))_i$. Note that $F(\boldsymbol{v})$ must satisfy the following constraints: (1) $(F(\boldsymbol{v}))_i \geq 0$ for all $i \in \{1, \ldots, Q\}$, (2) $\sum_{i=1}^Q (F(\boldsymbol{v}))_i = 1$. Thus, minimizing the prediction loss $\mathcal{L}_p(E, F)$ w.r.t. $F$ is equivalent to solve the following constrained optimization problem:

$$\begin{aligned}
\max_{F(\boldsymbol{v})} \quad & \sum_{i=1}^Q \mathbb{E}\left[y_i|\boldsymbol{v}\right] \log(F(\boldsymbol{v}))_i \\
\text{s.t.} \quad & \sum_{i=1}^Q (F(\boldsymbol{v}))_i = 1 \\
& (F(\boldsymbol{v}))_i \geq 0, \ i \in \{1, \ldots, Q\}.
\end{aligned}$$

To solve this constrained problem, we first define the Lagrangian function:

$$l(F(\boldsymbol{v}), \lambda, \boldsymbol{\mu}) = \sum_{i=1}^Q \mathbb{E}\left[y_i|\boldsymbol{v}\right] \cdot \log(F(\boldsymbol{v}))_i + \lambda\left(1 - \sum_{j=1}^Q (F(\boldsymbol{v}))_j\right) + \sum_{k=1}^Q \mu_k \cdot (F(\boldsymbol{v}))_k,$$

where $\lambda \geq 0$ and $\mu_i \geq 0$ for $i \in \{1, \ldots, Q\}$. By the first-order Karush-Kuhn-Tucker (KKT) conditions:

$$\begin{aligned}
\tfrac{\partial l}{\partial \lambda} &= 1 - \sum_{i=1}^Q (F(\boldsymbol{v}))_i = 0, \\
\tfrac{\partial l}{\partial F_i} &= \tfrac{\mathbb{E}[y_i|\boldsymbol{v}]}{(F(v))_i} - \lambda + \mu_i = 0, \ i \in \{1, \ldots, Q\}, \\
\tfrac{\partial l}{\partial \mu_i} &= (F(\boldsymbol{v}))_i \geq 0, \ i \in \{1, \ldots, Q\}, \\
\mu_i &\geq 0, \ i \in \{1, \ldots, Q\}, \\
\mu_i \cdot (F(\boldsymbol{v}))_i &= 0, \ i \in \{1, \ldots, Q\},
\end{aligned}$$

we can derive the optimal $(F(\boldsymbol{v}))_i$ for $i \in \{1, \ldots, Q\}$ as:

$$(F^*(\boldsymbol{v}))_i = \mathbb{E}\left[y_i|\boldsymbol{v}\right] = \mathbb{P}\left(y_i = 1|\boldsymbol{v}\right),$$

and

$$F^*(\boldsymbol{v}) = F^*(E(\boldsymbol{x})) = [\mathbb{P}(y_i = 1|\boldsymbol{v})]_{i=1}^Q \in \mathbb{R}^Q.$$

At that point, $\mathcal{L}_p(E, F)$ achieves its minimum value:

$$
\begin{aligned}
\mathcal{L}_p(E, F^*) &= \mathbb{E}\left[L_p(F^*(E(\boldsymbol{x})), \boldsymbol{y}\right] \\
&= \mathbb{E}_{\boldsymbol{v} \sim p(\boldsymbol{v})} \left[ \sum_{i=1}^Q \mathbb{E}\left[y_i|\boldsymbol{v}\right] \log \tfrac{1}{(F^*(\boldsymbol{v}))_i} \right] \\
&= \mathbb{E}_{\boldsymbol{v} \sim p(\boldsymbol{v})} \left[ \sum_{i=1}^Q \mathbb{P}(y_i = 1|\boldsymbol{v}) \log \tfrac{1}{\mathbb{P}(y_i=1|\boldsymbol{v})} \right] \\
&= H(\boldsymbol{y}|\boldsymbol{v}) = H(\boldsymbol{y}|E(\boldsymbol{x})),
\end{aligned}
$$

completing the proof. $\qquad\square$

**Theorem 4.2** (**Optimal Concept Embedding Encoder**). *Assuming $u \perp \boldsymbol{y}$, if the concept embedding encoder $E$, concept probability encoder $E_{prob}$, the predictor $F$ and the discriminator $D$ have enough capacity and are trained to reach optimum, any global optimal concept embedding encoder $E^*$ and its corresponding global optimal concept probability encoder $E^*_{prob}$ have the following properties:*

$$E^*_{prob}(\boldsymbol{x}) = [\mathbb{P}(c_i = 1|\boldsymbol{x})]_{i=1}^K, \tag{17}$$

$$H\left(\boldsymbol{y} \mid E^*(\boldsymbol{x})\right) = H(\boldsymbol{y} \mid \boldsymbol{x}), \tag{18}$$

$$\widetilde{C}_d\left(E^*\right) = \max_{E'} \widetilde{C}_d\left(E'\right). \tag{19}$$

*Proof.* We first prove the optimal concept probability encoder in Eqn. 17. Because $E_{prob}(\boldsymbol{x}) = [(E_{prob}(\boldsymbol{x}))_i]_{i=1}^K \in \mathbb{R}^K$, and $L_c$ is the average binary cross entropy:

$$L_c(E_{prob}(\boldsymbol{x}), \boldsymbol{c}) = \tfrac{1}{K} \sum_{i=1}^K c_i \log \tfrac{1}{(E_{prob}(\boldsymbol{x}))_i} + (1 - c_i) \log \tfrac{1}{1-(E_{prob}(\boldsymbol{x}))_i},$$

then we have

$$
\begin{aligned}
\mathbb{E}_S\left[L_c(E_{prob}(\boldsymbol{x}), \boldsymbol{c})\right] &= \tfrac{1}{K} \sum_{i=1}^K \mathbb{E}_S\left[ c_i \log \tfrac{1}{(E_{prob}(\boldsymbol{x}))_i} + (1 - c_i) \log \tfrac{1}{1-(E_{prob}(\boldsymbol{x}))_i} \right] \\
&= \tfrac{1}{K} \sum_{i=1}^K \mathbb{E}_{(\boldsymbol{x},\boldsymbol{c}) \sim p(\boldsymbol{x},\boldsymbol{c})}\left[ c_i \log \tfrac{1}{(E_{prob}(\boldsymbol{x}))_i} + (1 - c_i) \log \tfrac{1}{1-(E_{prob}(\boldsymbol{x}))_i} \right] \\
&= \tfrac{1}{K} \sum_{i=1}^K \mathbb{E}_{\boldsymbol{x} \sim \mathcal{D}_S}\left[ \mathbb{E}_{c_i \sim p(c_i|\boldsymbol{x})}\left[ c_i \log \tfrac{1}{(E_{prob}(\boldsymbol{x}))_i} + (1 - c_i) \log \tfrac{1}{1-(E_{prob}(\boldsymbol{x}))_i} \right] \right] \\
&= \tfrac{1}{K} \sum_{i=1}^K \mathbb{E}_{\boldsymbol{x} \sim \mathcal{D}_S}\left[ \mathbb{E}(c_i|\boldsymbol{x}) \log \tfrac{1}{(E_{prob}(\boldsymbol{x}))_i} + (1 - \mathbb{E}(c_i|\boldsymbol{x})) \log \tfrac{1}{1-(E_{prob}(\boldsymbol{x}))_i} \right].
\end{aligned}
$$

Thus, the optimal concept probability encoder $(E_{prob})_i$ for $i \in \{1, \ldots, K\}$ should be

$$
\begin{aligned}
(E^*_{prob})_i &= \underset{(E_{prob})_i}{\arg\min} \, \mathbb{E}_S\left[L_c(E_{prob}(\boldsymbol{x}), \boldsymbol{c})\right] \\
&= \underset{(E_{prob})_i}{\arg\min} \, \tfrac{1}{K} \sum_{j=1}^K \mathbb{E}_{\boldsymbol{x} \sim \mathcal{D}_S}\left[ \mathbb{E}(c_j|\boldsymbol{x}) \log \tfrac{1}{(E_{prob}(\boldsymbol{x}))_j} + (1 - \mathbb{E}(c_j|\boldsymbol{x})) \log \tfrac{1}{1-(E_{prob}(\boldsymbol{x}))_j} \right] \\
&= \underset{(E_{prob})_i}{\arg\min} \, \mathbb{E}_{\boldsymbol{x} \sim \mathcal{D}_S}\left[ \mathbb{E}(c_i|\boldsymbol{x}) \log \tfrac{1}{(E_{prob}(\boldsymbol{x}))_i} + (1 - \mathbb{E}(c_i|\boldsymbol{x})) \log \tfrac{1}{1-(E_{prob}(\boldsymbol{x}))_i} \right].
\end{aligned}
$$

For any $(a, b) \in \mathbb{R}^2 \backslash (0, 0)$, the function $y \to a \log(1 - y) + b \log(y)$ achieves its maximum in $[0, 1]$ at $\frac{b}{a+b}$. Applying this result, we derive the optimal value of $(E_{prob}(\boldsymbol{x}))_i$ for $i \in \{1, \ldots, K\}$ as:

$$(E_{prob}^*(\boldsymbol{x}))_i = \mathbb{E}\left(c_i | \boldsymbol{x}\right) = \mathbb{P}(c_i = 1 | \boldsymbol{x}),$$

and the optimal $E_{prob}(\boldsymbol{x})$ is given by

$$E_{prob}^*(\boldsymbol{x}) = [(E_{prob}^*(\boldsymbol{x}))_1, \ldots, (E_{prob}^*(\boldsymbol{x}))_K]^\mathsf{T}$$
$$= [\mathbb{P}(c_1 = 1 | \boldsymbol{x}), \ldots, \mathbb{P}(c_K = 1 | \boldsymbol{x})]^\mathsf{T},$$

completing the proof for Eqn. 17.

Since $E(\boldsymbol{x})$ is a function of $\boldsymbol{x}$, by the data processing inequality, we have

$$H(\boldsymbol{y} | E(\boldsymbol{x})) \geq H(\boldsymbol{y} | \boldsymbol{x}).$$

The objective function mentioned in Eqn. 16 has the following lower bound:

$$C(E) \triangleq H(\boldsymbol{y} \mid E(\boldsymbol{x})) - \lambda_d \widetilde{C}_d(E)$$
$$\geq H(\boldsymbol{y} \mid \boldsymbol{x}) - \lambda_d \max_{E'} \widetilde{C}_d(E').$$

This equality holds if and only if $H(\boldsymbol{y} \mid E(\boldsymbol{x})) = H(\boldsymbol{y} \mid \boldsymbol{x})$ and $\widetilde{C}_d(E) = \max_{E'} \widetilde{C}_d(E')$. Therefore, we only need to prove that the optimal value of $C(E)$ is equal to $H(\boldsymbol{y} \mid \boldsymbol{x}) - \lambda_d \max_{E'} \widetilde{C}_d(E')$ in order to prove that any global encoder $E^*$ satisfies Eqn. 18, and Eqn. 19.

We show that $C(E)$ can achieve its lower bound by considering the following encoder $E_0$: $E_0(\boldsymbol{x}) = P_{\boldsymbol{y}}(\cdot | \boldsymbol{x})$ (Zhao et al., 2017; Wang et al., 2020). It can be checked that $H(\boldsymbol{y} \mid E_0(\boldsymbol{x})) = H(\boldsymbol{y} \mid \boldsymbol{x})$ and $E_0(\boldsymbol{x}) \perp u$ which leads to $\widetilde{C}_d(E_0) = \max_{E'} \widetilde{C}_d(E')$, completing the proof. □

## C. Experiments

### C.1. Dataset Details

**Waterbirds Datasets (Sagawa et al., 2019).** First, we incorporate the concepts from the CUB (Wah et al., 2011) dataset into the original Waterbirds dataset to make it compatible with concept-based models. Since the original Waterbirds dataset is a binary classification task (landbirds are always associated with land and waterbirds with water as source domain), we construct the target domain, Waterbirds-shift (background shift data, the same construct method as CONDA (Choi et al., 2024) Waterbirds dataset), by selecting images with opposite attributes (e.g., landbirds in water and waterbirds on land). This results in Waterbirds-2, a binary classification domain adaptation dataset. Additionally, because the CUB dataset is inherently a multi-class classification task, we construct Waterbirds-200 by replacing the labels in the Waterbirds-2 dataset with the multi-class labels from CUB without modifying the data itself. Finally, as the CUB dataset represents a natural domain shift relative to Waterbirds-200, we use the CUB training data as the source domain and retain the Waterbirds-shift images as the target domain to construct Waterbirds-CUB.

**MNIST Concepts.** We selected 11 topology concepts *[Ring, Line, Arc, Corner, Top-Curve, Semicircles, Triangle, Bottom-Curve, Top-Line, Wedge, Bottom-Line]* (initially generated by GPT-4 (Achiam et al., 2023) and refined through manual screening) for the MNIST (LeCun et al., 1998), MNIST-M(Ganin et al., 2016), SVHN (Netzer et al., 2011), and USPS (Hull, 1994) digit datasets to evaluate the performance of our method. In addition, PCBMs (Yuksekgonul et al., 2022) can utilize the CLIP model (we tested with CLIP:RN50) (Radford et al., 2021) to automatically generate concepts. To evaluate its effectiveness, we compare the concepts generated by CLIP with our predefined set of concepts. However, since the PCBM-generated concepts are stored in a concept bank and lack explicit relationships between classes and concepts, they cannot be directly used to evaluate our model.

*Table 4.* Performance comparison across MNIST datasets using different concepts. The numbers 11 and 13 represent the concepts generated by PCBMs through once and twice recursive exploration of the ConceptNet (Speer et al., 2017) graph.

| Dataset | MNIST → MNIST-M | | | SVHN → MNIST | | | MNIST → USPS | | |
|---|---|---|---|---|---|---|---|---|---|
| Concepts | 11 | 13 | Ours | 11 | 13 | Ours | 11 | 13 | Ours |
| PCBM | $13.795_{\pm 0.549}$ | $11.906_{\pm 0.286}$ | $29.660_{\pm 1.020}$ | $11.350_{\pm 0.000}$ | $11.350_{\pm 0.000}$ | $21.323_{\pm 2.116}$ | $13.337_{\pm 0.183}$ | $14.117_{\pm 0.963}$ | $15.54_{\pm 0.115}$ |
| CONDA | $9.754_{\pm 0.000}$ | $9.754_{\pm 0.000}$ | $9.754_{\pm 0.000}$ | $9.800_{\pm 0.000}$ | $9.800_{\pm 0.000}$ | $9.800_{\pm 0.000}$ | $17.887_{\pm 0.000}$ | $17.887_{\pm 0.000}$ | $17.887_{\pm 0.000}$ |
| CUDA (Ours) | - | - | $\mathbf{95.24_{\pm 0.13}}$ | - | - | $\mathbf{82.49_{\pm 0.27}}$ | - | - | $\mathbf{96.01_{\pm 0.13}}$ |

**SkinCON Datasets (Daneshjou et al., 2022b).** The SkinCON dataset is constructed using two existing datasets: Fitzpatrick 17k (Groh et al., 2021) and Diverse Dermatology Images (DDI) (Daneshjou et al., 2022a). Both datasets are publicly available for scientific, non-commercial use. Fitzpatrick 17k, which was scraped from online atlases, contains a higher level of noise compared to DDI, making domain adaptation on Fitzpatrick 17k more challenging. However, due to the small size of the DDI dataset, we exclusively use Fitzpatrick 17k while excluding non-skin images (those with unknown skin tone types or labels not consider by SkinCON).

### C.2. Experimental Details

**Model and Optimization Details.** We use ResNet-50 (He et al., 2016) for the Waterbirds dataset and ResNet-18 for the MNIST and SkinCON datasets. The hyperparameters are summarized in Table 5. All DA baselines, CBMs (Koh et al., 2020), and CEMs (Zarlenga et al., 2022) share the same backbone as our approach for fair comparison. Zero-shot serves as the naive baseline for CONDA (Choi et al., 2024), where it uses the prompt "an image of [class]" to generate predictions. CONDA improves upon this by combining the zero-shot predictor with a linear-probing predictor to obtain pseudo-labels for the test batch, followed by test-time adaptation. Both PCBM and CONDA require pretrained models to construct the concept bank. Therefore, we utilize CLIP:ViT-L-14 (Radford et al., 2021) for the Waterbirds dataset consistent with CONDA, and CLIP:RN50 for the MNIST and SkinCON datasets. Our code will be available at https://github.com/xmed-lab/CUDA.

*Table 5.* Hyper-parameters of CUDA during training.

| | Leaning Rate | Weight Decay | $\lambda_c$ | $\lambda_d$ | Relax Threshold |
|---|---|---|---|---|---|
| Waterbirds-2 | $1e\text{-}3$ | $4e\text{-}5$ | 5 | 0.3 | 0.5 |
| Waterbirds-200/CUB | $1e\text{-}3$ | $4e\text{-}5$ | 5 | 0.3 | 0.7 |
| MNIST → MNIST-M/USPS | $1e\text{-}3$ | $1e\text{-}5$ | 5 | 0.1 | 0.6 |
| SVHN → MNIST | $1e\text{-}3$ | $1e\text{-}5$ | 5 | 0.1 | 0.7 |
| I-II → III-IV | $1e\text{-}3$ | $4e\text{-}5$ | 10 | 0.1 | 0.3 |
| III-IV → V-VI/I-II | $1e\text{-}3$ | $4e\text{-}5$ | 10 | 0.1 | 0.7 |

**Naive Baseline.** DA methods and the concept-driven paradigm of CBMs cannot be naively combined. In our naive baseline, we extend the DA model by adding a linear layer to its feature output layer to predict concepts, incorporating the concept loss into the original DA loss. However, as shown in Table 6, this approach fails to effectively capture concept information and performs worse than the original CBMs method. This highlights that the standard DA structure is not inherently suited for learning concepts and fails to leverage the benefits of concepts to improve domain alignment.

**Lipschitz Continuity.** Lemma 3.1 assumes that all hypotheses are $L$-Lipschitz continuous for some constant $L > 0$. While this assumption might seem restrictive at first glance, it is actually quite reasonable. In practice, hypotheses are often implemented using neural networks (e.g., our label predictor), where the fundamental components – such as linear layers and activation functions are naturally Lipschitz continuous. Therefore, this assumption is not overly strong and is typically satisfied (Shen et al., 2018).

### C.3. Framework Details and Training Algorithm

Our adversarial training process consists of two main steps. First, we optimize the domain discriminator using the original discriminator loss (Eqn. 8) and then calculate the relaxed discriminator loss (Eqn. 9). Second, we optimize the concept

*Table 6.* Performance of concept-based methods on both concept learning and classification across different datasets. CEM (w/o R.) indicates without RandInt. Naive refers our naive combination baseline. We mark the best result with **bold face** and the second best results with underline. Average accuracy is calculated over every three datasets of the same type images.

| Datasets | Waterbirds-2 | | | Waterbirds-200 | | | Waterbirds-CUB | | | AVG |
|---|---|---|---|---|---|---|---|---|---|---|
| Metrics | Concept | Concept F1 | Class | Concept | Concept F1 | Class | Concept | Concept F1 | Class | ACC |
| CEM | $94.14_{\pm 0.13}$ | $81.74_{\pm 0.39}$ | $70.27_{\pm 1.70}$ | $93.68_{\pm 0.10}$ | $81.22_{\pm 0.64}$ | $62.26_{\pm 1.11}$ | $93.64_{\pm 0.08}$ | $80.08_{\pm 0.34}$ | $66.48_{\pm 0.81}$ | 66.34 |
| CEM (w/o R.) | $94.17_{\pm 0.14}$ | $81.96_{\pm 0.30}$ | $69.45_{\pm 2.15}$ | $93.76_{\pm 0.20}$ | $81.04_{\pm 0.82}$ | $63.56_{\pm 1.25}$ | $93.66_{\pm 0.14}$ | $79.80_{\pm 0.36}$ | $65.89_{\pm 0.51}$ | 66.30 |
| CBM | $93.60_{\pm 0.20}$ | $83.89_{\pm 0.49}$ | $74.81_{\pm 2.16}$ | $93.50_{\pm 0.16}$ | $83.14_{\pm 0.98}$ | $63.89_{\pm 1.16}$ | $93.40_{\pm 0.14}$ | $82.10_{\pm 0.48}$ | $63.89_{\pm 1.00}$ | 67.53 |
| Naive | $85.41_{\pm 0.17}$ | $71.86_{\pm 0.19}$ | $66.83_{\pm 2.96}$ | $88.20_{\pm 0.04}$ | $73.96_{\pm 0.16}$ | $63.51_{\pm 0.32}$ | $88.11_{\pm 0.05}$ | $73.56_{\pm 0.09}$ | $60.72_{\pm 0.27}$ | 63.69 |
| **CUDA (Ours)** | **$94.63_{\pm 0.05}$** | **$84.97_{\pm 0.15}$** | **$92.90_{\pm 0.31}$** | **$95.15_{\pm 0.05}$** | **$85.06_{\pm 0.19}$** | **$75.87_{\pm 0.31}$** | **$94.58_{\pm 0.07}$** | **$82.81_{\pm 0.19}$** | **$74.66_{\pm 0.19}$** | **81.15** |

embedding encoder and label predictor. The overall framework is illustrated in Fig. 5, and the detailed training process is outlined in Algorithm 1. The objective is to learn both the labels and concepts in the target domain, given source and target domain images as input. The training procedure alternates between Eqn. 4 and 5 with adversarial training using Eqn. 6∼9. During inference, we predict the target domain class label $\widehat{y} = F(E(\boldsymbol{x}))$ and concepts $\widehat{c} = E_{prob}(\boldsymbol{x})$. The code will be released upon the acceptance of this work.

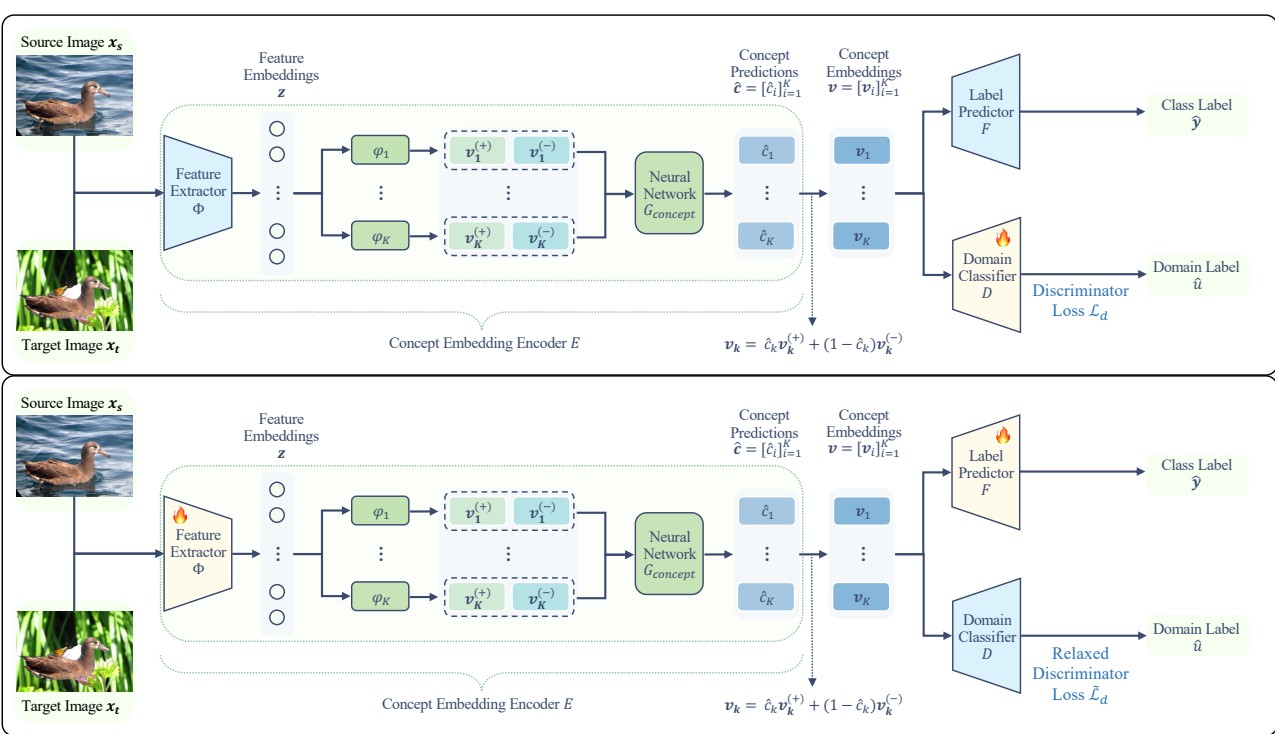

*Figure 5.* The full CUDA framework. It processes source and target domain images to learn feature embeddings, from which positive $\boldsymbol{v}_i^{(+)}$ and negative $\boldsymbol{v}_i^{(-)}$ embeddings are derived. These embeddings are passed through $G_{\text{concept}}$ to compute concept predictions $\widehat{c}$ and construct final concept embeddings $\boldsymbol{v}$. Adversarial training alternates between optimizing the domain classifier (discriminator) with Eqn. 4 and optimizing the concept embedding encoder and label predictor with Eqn. 5, guided by adversarial training using Eqn. 6∼9.

## D. Limitations and Future Works

Our approach falls short of the state-of-the-art UDA method GH++ (Huang et al., 2024) on the Waterbirds 200 classification task. This may be attributed to GH++'s use of gradient harmonization, which balances the classification and domain alignment tasks, particularly benefiting scenarios with a large number of categories. Exploring how to leverage gradient harmonization to balance concept learning in our framework is an interesting direction for future work. We plan to

investigate the related theoretical foundations and explore how it can be effectively integrated into our method in future work. Additionally, our approach achieves competitive results and stable performance without using the most advanced DA backbone. We believe that plugging our method into a more sophisticated backbone could lead to even more remarkable performance, which we leave as future work. Lastly, the domain shift studied in this work primarily involves shifts related to background or other label-agnostic factors. In the future, we aim to extend our method to address other types of domain shifts, broadening its applicability to other scenarios.

---

**Algorithm 1** Pseudocode of `CUDA` Training

---

**Input:** Source domain data $\mathcal{S} = \{(\boldsymbol{x}_i^s, \boldsymbol{y}_i^s, \boldsymbol{c}_i^s)\}_{i=1}^n$, target domain data $\mathcal{T} = \{\boldsymbol{x}_j^t\}_{j=1}^m$, feature extractor $\Phi$, concept embedding generator $\varphi$, concept probability network $G_{\text{concept}}$, label predictor $F$, domain discriminator $D$, learning rates $\alpha_1$, $\alpha_2$, concept loss weight $\lambda_c$, domain discriminator loss weight $\lambda_d$, concept number $K$, relaxation threshold $\tau$.
**Output:** Predicted target labels and concepts $\{\widehat{\boldsymbol{y}}_t, \widehat{\boldsymbol{c}}_t\}$.

1: **while** not converged **do**
2:      Sample minibatches $\mathcal{X}_s \subset \mathcal{S}$ and $\mathcal{X}_t \subset \mathcal{T}$.
3:      **for** each domain $d \in \{s, t\}$ (*source* or *target*) **do**
4:          Extract feature embeddings: $\boldsymbol{z}_d \leftarrow \Phi(\boldsymbol{x}_d)$.
5:          **for** $k = 1$ to $K$ **do**
6:              Generate positive and negative concept embeddings: $[\boldsymbol{v}_{d,k}^{(+)}, \boldsymbol{v}_{d,k}^{(-)}] \leftarrow \varphi_k(\boldsymbol{z}_d)$.
7:              Predict concept probabilities:
                  $\widehat{c}_{d,k} \leftarrow G_{\text{concept}}([\boldsymbol{v}_{d,k}^{(+)}, \boldsymbol{v}_{d,k}^{(-)}])$.
8:              Combine positive and negative embeddings:
                  $\boldsymbol{v}_{d,k} \leftarrow \widehat{c}_{d,k} \cdot \boldsymbol{v}_{d,k}^{(+)} + (1 - \widehat{c}_{d,k}) \cdot \boldsymbol{v}_{d,k}^{(-)}$.
9:          **end for**
10:     **end for**
11:     Predict source class labels: $\widehat{\boldsymbol{y}}_s \leftarrow F(\boldsymbol{v}_s)$.
12:     Predict domain labels: $\widehat{u}_s \leftarrow D(\boldsymbol{v}_s), \widehat{u}_t \leftarrow D(\boldsymbol{v}_t)$.
13:     Compute $\mathcal{L}_p(\widehat{\boldsymbol{y}}_s, \boldsymbol{y}_s)$ based on Eqn. 6.
14:     Compute $\mathcal{L}_c(\widehat{\boldsymbol{c}}_s, \boldsymbol{c}_s)$ based on Eqn. 7.
15:     Compute $\mathcal{L}_d(\widehat{u}_\theta, u_\theta), \theta \in \{s, t\}$ based on Eqn. 8.
16:     Relax the domain discriminator loss to get $\widetilde{\mathcal{L}}_d$ based on Eqn. 9.
17:     $\mathcal{L}_{total} \leftarrow \mathcal{L}_p + \lambda_c \mathcal{L}_c - \lambda_d \widetilde{\mathcal{L}}_d$
18:     Update $D$ to minimize $\mathcal{L}_d$ with learning rate $\alpha_1$.
19:     Update $\Phi, \varphi, G_{\text{concept}},$ and $F$ to minimize $\mathcal{L}_{total}$ with learning rate $\alpha_2$.
20: **end while**
21: **return** $\{\widehat{\boldsymbol{y}}_t, \widehat{\boldsymbol{c}}_t\}$

---

