# OpenReview forum: "Concept-Based Unsupervised Domain Adaptation"
_ICML.cc/2025/Conference — ICML 2025 poster_

### Official Review · Reviewer_sn2y · 2025-03-13

**Overall Recommendation:** 3

**Summary:**

This paper aims to improve the generalizability and transferability of Concept Bottleneck Models (CBMs) by proposing a novel Concept-based Unsupervised Adaptation (CUDA) framework. The CUDA framework is designed to align concepts across domains in an adversarial manner while introducing a relaxation threshold. The authors evaluated the proposed framework across three sets of domain adaptation datasets.

**Claims And Evidence:**

While the authors have put in good effort in proving the generalization error bound for CBMs with the concept terms and also the optimality of the designed CUDA framework with the relaxed discriminator loss, it can be observed that the proves are discussed within the concept space which is assumed to be a separate space than that of the original feature space. This allows Theorem 3.1 over the target-domain error to be defined directly for the source and target concept spaces. However, the ceoncept embeddings originates from the feature embeddings which is part of the concept embedding encoder. Could the authors explain the relationship between the target domain error that is bound for the CBM (Formula 3) and the original target domain error (changing $\tilde{D}_S$ and $\tilde{D}_T$ in Formula 1 with $D_S$ and $D_T$)? That would help to understand how different the target error will be in the context of feature embedding and concept embedding.

**Essential References Not Discussed:**

Not available (to the best of my knowledge)

**Experimental Designs Or Analyses:**

One outstanding result that can be observed is the efficacy of CUDA on WB-2, where the second best result is an astonishingly 20+ percent lower than that of CUDA while the majority methods (even methods such as GH++ which performs well on WB-200 and WB-CUB) scores lower than 50, which indicates the performance to be worse than even a wild guess. Could the authors explain the huge gap between the results and why CUDA stood out much more in WB-2 than other WB related datasets?

**Methods And Evaluation Criteria:**

The authors evaluate the methds against a number of CBMs and UDA methods on three sets of benchmarks. However, some widely used DA benchmarks, including DomainNet, VisDA-2017, and Office-31 (or Office-Home/Office-Caltech) have not been leveraged. Therefore while the efficacy of CUDA is indeed proven against prior CBMs and some very prior UDA methods (DANN, MCD), its efficacy in more comprehensive benchmarks against more recent UDA methods are not sufficiently proven.

**Other Comments Or Suggestions:**

Not available, see above for details.

**Other Strengths And Weaknesses:**

Overall, the paper is well written with comprehensive proofs and sufficient results. The idea of combining DA with CBM is interesting and could potentially be leveraged for improving both the generalizability and interpretability of current methods. However, there are some concerns over the assumptions that leads to the computation of the error upper bound with the concept embedding, and the evaluation benchmarks, highlighted as above.

**Questions For Authors:**

Not available, see above for details.

**Relation To Broader Scientific Literature:**

The paper touches upon domain adaptation in the context of CBM, which relates to the domain adaptation in the context of feature spaces.

**Theoretical Claims:**

The proofs provided are mostly detailed and correct.

---

> ### Author Rebuttal · Authors · 2025-03-30
>
> Thank you for your valuable comments. We are glad that you found our framework ``"novel"``, our idea ``"interesting"``, and our proofs ``"correct"``. We address your comments in turn below.
>
> **Q1. ... relationship between the target domain error ... CBM (Formula 3) and the original target domain error in feature space ...?**
>
> To see the relationship between Formula 3 and the original target domain error in feature space, we start with the target error defined in the feature space (denoted as $\\mathcal{Z}$):
>
> **Without concept learning**, we have the standard domain adaptation bound as (similar to Formula 1):
>
> $$
> \\epsilon_{T}^{\\boldsymbol{z}}(h) \\leq \\epsilon^{\\boldsymbol{z}}_{S}(h) + \\frac{1}{2} d _{\\mathcal{H} \Delta \\mathcal{H}}(\\mathcal{D}^{\\boldsymbol{z}} _S, \\mathcal{D}^{\\boldsymbol{z}} _T) + \\eta^{\\boldsymbol{z}},
> $$
>
> where $\mathcal{D}_S^{\boldsymbol{z}}$ and  $\mathcal{D}_T^{\boldsymbol{z}}$ are denoted as the induced marginalized distributions in $\mathcal{Z}$, and all terms are defined in this space.
>
> **With concept learning**, the bound above can be refined to include complex transformation error terms (resembling Formula 3):
>
> $$
> \\epsilon\_T^{\\boldsymbol{z}}(h) \\leq \\epsilon^{\\boldsymbol{z}} _S(h) + \\frac{1}{2} d _{\\mathcal{H}\\Delta\\mathcal{H}}(\\mathcal{D}^{\\boldsymbol{z}} _S, \\mathcal{D}^{\\boldsymbol{z}} _T) + ( const \times \\mathbb{E} _S [ || \\hat{\\boldsymbol{c}} - \\boldsymbol{c}|| ] + error _{other}) + \\eta^{\\boldsymbol{z}},
> $$
>
> where $error _{other}$ contains additional losses during the transformation from $\\mathcal{Z}$ to $\\mathcal{V}$ (e.g., projection errors). If $error _{other}$ is spread and absorbed into $\\epsilon _S^{\\boldsymbol{z}}(h)$, $d _{\\mathcal{H} \\Delta \\mathcal{H}}( \\mathcal{D} _S^{\\boldsymbol{z}}, \\mathcal{D} _T^{\\boldsymbol{z}})$, and $\\eta^{\\boldsymbol{z}}$, the resulting bound becomes structurally equivalent to Formula 3 (copied below):
>
> $$
> \\epsilon _{T}(h)  \\leq \\epsilon _{S}^{\\boldsymbol{c}}(h) + \\frac{1}{2} d _{\\mathcal{H} \\Delta \\mathcal{H}}(\\tilde{\\mathcal{D}}^{\\boldsymbol{c}}_S, \\tilde{\\mathcal{D}} _T)  + R \\cdot \\mathbb{E} _{S} [ || \\hat{\\boldsymbol{c}} - \\boldsymbol{c} ||] + \\eta^{\\boldsymbol{c}}.
> $$
>
> Thus, Formula 3 can be viewed as a refinement of the feature-space bound, making concept prediction error explicit and improving interpretability.
>
> **Q2. ... widely used DA benchmarks ... against more recent UDA methods ...**
>
> Good question.
>
> **Mainstream DA Datasets Are Not Applicable:** Mainstream DA datasets for DA such as Office-31, Office-home, VisDA-2017 are not applicable to our setting because they do not contain concept annotations, which are needed to evaluate the correctness of concept prediction (and interpretation) in all methods.
>
> **Additional Datasets:** As you suggested, we add a new dataset: AwA2-ImageNet, where AwA2 [Liu et al. 2015] is the source domain containing 85 concepts for animals, and ImageNet is the target domain. We match the AwA2 classes with corresponding subsets of ImageNet and filter overlapping data. We further perform style transfer on ImageNet to induce domain shift, making the task more challenging. The table below summarizes the results, which demonstrate our CUDA outperforms existing CBM variants and DA methods even in the case of larger domain gaps and more diverse data.
>
> |Metrics|Concept Accuracy|Concept F1|Class Accuracy|
> | :-----: | :--------: | :--------: | :--------: |
> |CBM|80.15±0.09|59.25±0.64|33.89±1.12|
> |CEM|80.63±0.37|59.17±0.19|33.73±2.81|
> |GH++|-|-|54.83±0.23|
> |CUDA|86.84±0.30| 65.60±0.26 |55.30±0.11|
>
> **More Recent UDA methods.** We did compare CUDA with GH++, a more recent, state-of-the-art UDA method (PAMI '24) that outperforms DANN and MCD. Our results show CUDA
> + enables concept interpretation while GH++ cannot, and
> + achieves competitive prediction performance.
> We also follow your suggestion to include more DA baselines, GVB (CVPR '20) and DWL (CVPR '21); see our **response to Q2&W2 for Reviewer tPTT**.
>
> **Q3. ... why CUDA stood out much more in WB-2 than other WB related datasets?**
>
> **Poor Performance of DA Methods on WB-2.** DA methods, such as GH++, rely heavily on class label information during training, without access to concept annotation. WB-2 has only binary labels, which contain limited information; it therefore leads to worse performance for DA methods.
>
> **Large Improvement from CUDA in WB-2.** In contrast, CUDA utilizes concept learning and relaxed alignment, effectively leveraging the 112 concepts for binary classification in WB-2.
>
> **Smaller Improvement in WB-200.** However, when applied to WB-200 or WB-CUB, it is more challenging for 112 concepts to represent 200 classes, leading to CUDA's smaller relative advantage against baselines.
>
> **Q4. Code reproducibility.**
>
> Thank you for your interest. We have finished cleaning up the source code and will release it for reproducibility.

---

### Official Review · Reviewer_1Zod · 2025-03-13

**Overall Recommendation:** 3

**Summary:**

This paper introduces a framework to handle domain shifts effectively while maintaining interpretability. It leverages adversarial training and a relaxed alignment mechanism to align concept embeddings across source and target domains, allowing for flexibility in capturing domain-specific variations. It also eliminates the need for labeled concept data in the target domain, enabling it to operate in unsupervised settings. The framework is supported by theoretical guarantees, including a generalization error bound and analysis of relaxed alignment, and achieves state-of-the-art performance in improving both concept accuracy and classification accuracy. Experiments on benchmark datasets demonstrate its significant improvements.

**Claims And Evidence:**

1. Theoretical analysis and experimental results show that relaxed alignment achieves better concept distribution matching and higher target accuracy.
2. Self-supervised learning and adversarial training are used to infer concepts in the target domain without explicit labels, validated by experiments across multiple datasets.
3. Figure 3 shows that the proposed method achieves better performance with higher intervention ratios compared to baseline methods.

**Essential References Not Discussed:**

No critical omissions

**Experimental Designs Or Analyses:**

1. The experiments cover a wide range of datasets and scenarios, including background shifts and medical imaging, which enhance the framework’s applicability.

**Methods And Evaluation Criteria:**

1. The proposed methods are well-tailored to address domain adaptation and concept learning challenges.
2. The use of diverse datasets strengthens the evaluation’s comprehensiveness.

**Other Comments Or Suggestions:**

None

**Other Strengths And Weaknesses:**

Strengths:
1. The introduction of relaxed alignment for concept-based domain adaptation is novel and impactful.
2. The method addresses a critical gap in interpretable domain adaptation, enhancing both accuracy and interpretability.
3. The paper is well-organized.

Weakness:
1. The paper lacks visual or qualitative examples to illustrate how relaxed alignment impacts interpretability.

**Questions For Authors:**

Can the proposed method scale to larger datasets or more complex domains? Have you tested this?

**Relation To Broader Scientific Literature:**

The paper builds upon works like CBMs, CEMs and DANN. It extends these by integrating interpretability into domain adaptation.

**Theoretical Claims:**

The proofs for Theorem 3.1 and Theorem 4.1 were reviewed. They are rigorous and mathematically sound.

---

> ### Author Rebuttal · Authors · 2025-03-30
>
> Thank you for your valuable comments. We are glad that you found our method ``"novel and impactful"``, our theoretical guarantees ``"rigorous and mathematically sound"``, and the experiments ``"comprehensive"`` and ``"applicable across a wide range of scenarios"``. Below, we address your comments one by one.
>
>
> **W1. The paper lacks visual or qualitative examples to illustrate how relaxed alignment impacts interpretability.**
>
> Thanks for mentioning this. Following your suggestion, we provide a more concrete example to illustrate how relaxed alignment impacts interpretability. We will also include this example into our revised version.
>
> We consider the task of making predictions for a target-domain image of a Black-Footed Albatross. The table below summarizes the predictions of different methods for both concept predictions and class label predictions; note that "CUDA w/o Relax" means "CUDA w/o Relaxed Alignment". Each row corresponds to one method, while the columns represent the predictions for specific concepts and class labels.
>
>
> + **Concept Predictions:** The first three columns show predictions for specific concepts (e.g., Seabird Bill, Black Nepe, Solid Breast). The values "0" and "1" indicate whether the prediction is incorrect or correct, respectively. The ground-truth distribution (GT) of each concept (in terms of GT positive rates) in the source and target domains is provided in parentheses ("GT Ratio: Source / Target").
>
> + **Class Label Predictions:** The last two columns show the predicted and ground-truth labels indices for each method.
>
>
> |      Method      | Seabird Bill (6.10% / 5.90%) | Black Nepe (48.59% / 50.69%) | Solid Breast (64.72% / 64.55%) | Predicted Label Index | Ground-Truth Label Index |
> | :--------------: | :--------------------------: | :--------------------------: | :----------------------------: | :-------------------: | :----------------------: |
> |       CBM        |              0               |              0               |               1                |          24           |            0             |
> |       CEM        |              1               |              0               |               1                |           0           |            0             |
> | CUDA (w/o Relax) |              0               |              1               |               1                |           0           |            0             |
> |     **CUDA**     |              1               |              1               |               1                |           0           |            0             |
>
>
>
> We can see that:
>
> + CBM predicts incorrect concepts, resulting in an incorrect class label prediction.
> + CEM and CUDA (w/o Relax.) predict the correct class label but fail to predict some concepts correctly, which harms interpretability.
> + Our full method (CUDA with Relaxed Alignment) predicts both the concepts and the class label correctly in the target domain, demonstrating that it achieves both interpretability and performance simultaneously.
> + Interestingly, the table shows that concepts with clear distribution differences (e.g., "Seabird Bill" and "Black Nepe") lead to incorrect concept predictions in models without the relaxed alignment mechanism (e.g., CUDA w/o Relax. and CBM predict "Seabird Bill" incorrectly). This further supports the importance of our relaxed alignment.
>
>
> **Q1. Can the proposed method scale to larger datasets or more complex domains? Have you tested this?**
>
> Yes, our method can scale to larger datasets and more complex domains. To demonstrate this, we followed your suggestion to conduct additional experiments on ImageNet. In particular, we include a new dataset: AwA2-ImageNet [4], where AwA2 is the source domain containing 85 concepts for animals, and ImageNet is the target domain. We match the AwA2 classes with corresponding subsets of ImageNet and filter overlapping data. Furthermore, we perform style transfer on the target domain (ImageNet) to induce domain shift, making the task more challenging. The table below summarizes the results, which
> + demonstrate our CUDA can improve performance even in the case of larger domain gaps and more diverse data and
> + highlight the scalability of our approach while maintaining its performance and interpretability in larger and more complex domains.
>
>
> | Metrics | Concept Accuracy |    Concept F1    |   Class Accuracy   |
> | :-----: | :--------: | :--------: | :--------: |
> |   CBM   | 80.15±0.09 | 59.25±0.64 | 33.89±1.12 |
> |   CEM   | 80.63±0.37 | 59.17±0.19 | 33.73±2.81 |
> |  GH++   |     -      |     -      | 54.83±0.23 |
> |  CUDA   | 86.84±0.30 | 65.60±0.26 | 55.30±0.11 |
>
>
> [4] Deep learning face attributes in the wild, CVPR15.

---

### Official Review · Reviewer_tPTT · 2025-03-14

**Overall Recommendation:** 3

**Summary:**

This paper aims to tackle the problem of limited generalization of concept bottleneck models in cross-domain scenarios. It utilizes adversarial training to align the cross-domain concept embeddings and introduces a relaxed uniform alignment technique to alleviate the influence of over-restricted domain alignment assumptions. Experiments on diverse datasets validate its effectiveness.

**Claims And Evidence:**

Yes, the author has validated his hypotheses and claims through clear and convincing evidence, including extensive theoretical derivation and proof, as well as experimental results.

**Essential References Not Discussed:**

To my knowledge, the paper includes all the necessary related works essential for its understanding with discussion. But there are several UDA methods are not included in the experiment section for quantitative comparison, such as [1-3] listed below.

**Experimental Designs Or Analyses:**

The experimental section shows the comparison with other CBM and DA methods, by which I check the validity of the designs and analyses.

**Methods And Evaluation Criteria:**

Yes, the methods CUDA and evaluation criteria for concept and class prediction make sense for the problem.

**Other Comments Or Suggestions:**

The paper is well-written and easy to follow, I have no further comments here.

**Other Strengths And Weaknesses:**

1.	For the over-constraints for domain distributions alignment, can the author provide some concrete examples for that, source and target concepts indeed have minor distribution differences?
2.	In experiments, the author includes several datasets including bird, digital and medical scenarios. But several mainstream datasets for DA are not considered, such as Office-31, Office-home, VisDA-2017, etc.

**Questions For Authors:**

1.	The author is suggested to provide more concrete examples to convince me the existence of concept distribution differences between source and target domains, and such discrepancy is a common problem in diverse DA scenarios, to make the questions this paper claimed for DA task more valuable.
2.	The author should include the experimental results on more datasets as mentioned in weakness, and compare with more UDA methods such as [1-3].


[1] Gradually vanishing bridge for adversarial domain adaptation, CVPR20
[2] Dynamic weighted learning for unsupervised domain adaptation, CVPR21
[3] Safe self-refinement for transformer-based domain adaptation, CVPR22

**Relation To Broader Scientific Literature:**

The introduced adversarial training and relaxed uniform alignment technique may be extended toward multi-class classification scenarios, but the author leaves it for future research.

**Theoretical Claims:**

I check the correctness of the proof of Optimal Discriminator(Lemma 4.1, B.2 in supplementary material), I believe this proof is correct and there are no issues with it.

---

> ### Author Rebuttal · Authors · 2025-03-30
>
> Thank you for your valuable comments. We are glad that you found our theory ``"extensive"``/``"correct"``, our claim ``"convincing"``, our experimental designs/analyses ``"valid"``, and the paper ``"well-written"``. Below, we address your questions one by one.
>
>
> **W1 & Q1. Provide more concrete examples to demonstrate concept distribution differences between source and target domains.**
>
> + One example is the concept "Primary Color: Brown", where the positive rates are 19% and 17% for the source and target domains, respectively. Additional examples include "Black Bill" (48% in source / 51% in target) and "All-purpose Bill" (40% in source / 41% in target).
>
> + Concept distribution differences are indeed a common problem in domain adaptation (DA) datasets with concept annotations. By directly counting the dataset statistics, we observe such discrepancies frequently arise across diverse DA scenarios.
>
>
> **W2 & Q2. Include experimental results on more datasets and compare with additional UDA methods (e.g., [1-3]).**
>
> This is a good question.
>
> **Common Datasets for Concept Bottleneck Models:** Since our CUDA belongs to the category of concept bottleneck models (CBMs), we follow the literature (e.g., CBMs [1], CEMs [2], and PCBMs [3]) to use common datasets such as CUB, MNIST, and SkinCON in our experiments, as they provide rich concept annotations essential for evaluating concept-based methods.
>
> **Mainstream Datasets for DA Are Not Applicable:** Note that mainstream datasets for DA such as Office-31, Office-home, VisDA-2017 are not applicable because they do not contain concept annotations, which are needed to evaluate the correctness of concept prediction in all methods.
>
>
>
> [1] Koh et al., ICML'20. [2] Zarlenga et al., NIPS'22. [3] Yuksekgonul et al., ICLR'22.
>
> **Additional Datasets:** Inspired by your comments, we add a new dataset: AwA2-ImageNet [4], where AwA2 is the source domain containing 85 concepts for animals and ImageNet is the target domain. We match the AwA2 classes with corresponding subsets of ImageNet and filter overlapping data.  Furthermore, we perform style transfer on the target domain (ImageNet) to induce domain shift, making the task more challenging. The table below summarizes the results, which demonstrate our CUDA can improve performance even in the case of larger domain gaps and more diverse data.
>
>
> | Metrics | Concept Accuracy |    Concept F1    |   Class Accuracy   |
> | :-----: | :--------: | :--------: | :--------: |
> |   CBM   | 80.15±0.09 | 59.25±0.64 | 33.89±1.12 |
> |   CEM   | 80.63±0.37 | 59.17±0.19 | 33.73±2.81 |
> |  GH++   |     -      |     -      | 54.83±0.23 |
> |  CUDA   | 86.84±0.30 | 65.60±0.26 | 55.30±0.11 |
>
>
> [4] Deep learning face attributes in the wild, CVPR'15.
>
> **Comparison with UDA Methods [1-3]:** Following your suggestion, we tested [5] and [6] on our nine datasets, and the results are summarized in the table below. However, [7] relies on a ViT backbone, making it unsuitable for fair comparison with our results, which use ResNet50. We agree that integrating transformer-based methods into our framework is a promising direction for future work. We will include these references in our paper. Thank you for this insightful suggestion!
>
> | Datasets |    WB-2    |   WB-200   |   WB-CUB   |  AVG  |   M → M-M  |    S → M   |    M → U   |  AVG  | I-II → III-IV | III-IV → V-VI | III-IV → I-II |  AVG  |
> |:------:|:----------:|:----------:|:----------:|:-----:|:----------:|:----------:|:----------:|:-----:|:-------------:|:-------------:|:-------------:|:-----:|
> |   GVB  | 40.97±3.78 | 74.89±0.69 | 73.66±0.35 | 63.17 | 41.28±3.55 | 81.21±0.36 | 94.27±1.64 | 72.25 |   76.35±0.73  |   78.3±1.43   |   74.32±0.1   | 76.32 |
> |   DWL  | 54.86±2.07 | 73.64±0.38 | 72.31±0.52 | 66.94 | 50.69±3.25 | 83.97±2.19 | 96.29±0.13 | 76.98 |   74.98±0.72  |   79.24±0.83  |   73.97±0.97  | 76.06 |
> |  CUDA  | 92.90±0.31 | 75.87±0.31 | 74.66±0.19 | 81.15 | 95.24±0.13 | 82.49±0.27 | 96.01±0.13 | 91.25 |   78.85±0.31  |   80.58±0.72  |   76.53±0.49  | 78.65 |
>
> [5] Gradually vanishing bridge for adversarial domain adaptation, CVPR20.
>
> [6] Dynamic weighted learning for unsupervised domain adaptation, CVPR21.
>
> [7] Safe self-refinement for transformer-based domain adaptation, CVPR22.

---

### Decision · Program_Chairs · 2025-05-01

**Decision:**

Accept (poster)

**Comment:**

This paper makes a valuable and timely contribution by connecting concept-based interpretability with domain adaptation—a direction that has been underexplored. The relaxed alignment mechanism is well-articulated, and both theoretical and empirical sections are solid. Reviewers raised valid concerns about experimental breadth and interpretability demonstrations, but these were mostly addressed during the rebuttal. On balance, this work appears to be a step forward in interpretable domain adaptation, with sound methodology and promising results. While some extensions could enhance the paper further, its core ideas are clear, technically correct, and of interest to the ICML community. I recommend acceptance.